

# Adaptively monitoring streamflow using a stereo computer vision system

Nicholas R. Hutley[1], Ryan Beecroft[1], Daniel Wagenaar[2], Josh Soutar[3], Blake Edwards[4], Nathaniel Deering[1], Alistair Grinham[1], Simon Albert[1]

[1]School of Civil Engineering, The University of Queensland, Brisbane, 4072, Australia
[2]Xylem Water Solutions, Newcastle, 2292, Australia
[3]Xylem Water Solutions, Brisbane, 4174, Australia
[4]Leading Edge Engineering Solutions, Albury, 2640, Australia

*Correspondence to*: Nicholas R. Hutley (nicholas.hutley@uq.net.au)

**Abstract.** The gauging of free surface flows in waterways provides the foundation for monitoring and managing the water resources of built and natural environments. A significant body of literature exists around the techniques and benefits of optical surface velocimetry methods to estimate flows in waterways without intrusive instruments or structures. However, to date the operational application of these surface velocimetry methods has been limited by site configuration and inherent challenging optical variability across different natural and constructed waterway environments. This work demonstrates a significant

advancement in the operationalisation of non-contact stream discharge gauging applied in the computer vision stream gauging (CVSG) system through the use of methods for remotely estimating water levels and adaptively learning discharge ratings over time. A cost-effective stereo camera-based stream gauging device (CVSG device) has been developed for streamlined site deployments and automated data collection. Evaluations between reference state-of-the-art discharge measurement technologies using DischargeLab (using surface structure image velocimetry), Hydro-STIV (using space-time image

velocimetry), ADCPs (acoustic doppler current profilers), and gauging station discharge ratings demonstrated that the optical surface velocimetry methods were capable of estimating discharge within best available measurement error margins of 5-15%. Furthermore, results indicated model machine learning approaches leveraging data to improve performance over a period of months at the study sites produced a marked 5-10% improvement in discharge estimates, despite underlying noise in stereophotogrammetry water level or optical flow measurements. The operationalisation of optical surface velocimetry

technology, such as CVSG, offers substantial advantages towards not only improving the overall density and availability of data used in stream gauging, but also providing a safe and non-contact approach for effectively measuring high flow rates while providing an adaptive solution for gauging streams with non-stationary characteristics.

## 1 Introduction

Globally, hydrological flow occurs through natural and man-made open channels and floodplains, often transporting life-

sustaining water to ecosystems and civilisations (Herrera et al., 2017; Albert et al., 2017; Grinham, 2007; Prüss-Ustün et al.,



2014; Albert et al., 2021). Likewise, rainfall variability with increasing risk due to climate change, can cause extreme flows (Lehmann et al., 2015; Palmer and Räisänen, 2002) resulting in significant economic, environmental, and life losses (Gaume et al., 2009; Grinham et al., 2012), as well as an increasing risk of extreme drought events into the future (Park et al., 2021; Li et al., 2016). Within this context the field of hydrography endeavours to monitor and understand the dynamics of flows in these

waterways through space and time (Westerberg et al., 2016; Kuentz et al., 2017; Mcmillan et al., 2012). From designing infrastructure intersecting with waterways (Lindow and Curtis, 2010), to budgeting water security (Daly et al., 2019; Sene et al., 2018) and improving forecasting models for near and long-term policy planning (Hering et al., 2015; Hutley et al., 2020), gauging waterways continues to be an important utility for society with substantial time, human risk, investment and maintenance funding worldwide (Crochemore et al., 2020).

There are many operational and emerging methods for waterway gauging, varying widely in cost, accuracy, reliability, and risk (Tauro et al., 2018; Gordon, 1989; Costa et al., 2000; Tauro et al., 2016; Yang et al., 2020). Intrusive methods range from the resource intensive installation of hydraulic control structures to measure discharge rates analytically using simpler water level measurements within a designed range by obstructing and controlling the flow through a standardised geometry (Boiten, 2002) (often to the detriment of aquatic species (Mueller et al., 2011), as well as sedimentation and erosion (Pagliara and

Palermo, 2015; Ogden et al., 2011)), through to the risking of people and equipment entering the stream to measure velocities using passive mechanical current meters or active acoustic doppler velocimetry profiles (Gordon, 1989).

In order to estimate discharges through a waterway without the flow passing through the geometry of a known hydraulic control structure, other methods largely rely on the measurement of velocities across the channel and integrating these estimated velocities through the cross-section area using some binned resolution (Herschy, 1993). Measuring these velocities

within the cross-section of waterways is fraught with various challenges, including limitations in measuring close to boundaries, debris, vegetation, aeration, unsteady flows, equipment damage, and safety risks (Petrie et al., 2013; Lee et al., 2014; Klema et al., 2020; Harding et al., 2016). Furthermore, the ongoing measurement of velocities in a waterway are difficult and expensive to carry out and maintain, especially during flood events (Banasiak and Krzyżanowski, 2015). Therefore, the development of discharge ratings (relating water level to an estimated discharge) have become common-place through the

construction of (often still expensive to build and maintain) gauging structures and stilling wells. This approach allows the more easily measured water level over time to be converted to discharge estimates through a fitting of manual discharge measurements recorded routinely and/or opportunistically over decades by professional hydrographers.

In practice, the common approach to gauging streams using fitted discharge ratings presents challenges for obtaining unbiased measurements of non-stationary channel environments over time, particularly when significant flow events cause changes to

natural waterways (Birgand et al., 2013; Tomkins, 2014; Guerrero et al., 2012; Jalbert et al., 2011; Di Baldassarre and Montanari, 2009). However, similar to one of the oldest manual methods to measure velocities in a waterway by measuring the displacements of surface floats over time, the passive optical measurement of surface velocities using relatively inexpensive camera systems has been an attractive approach to stream gauging (Dobriyal et al., 2017). Despite the documented advantages of this approach (Dramais et al., 2011) coupled with its potential for affordable scalability to decrease monitoring sparsity,



traditional gauging approaches have not yet been replaced or augmented by a widespread adoption of optical surface velocimetry after 20 years of active research (Tauro et al., 2018). Optical surface velocimetry in the form of large-scale particle image velocimetry (LSPIV) and space-time image velocimetry (STIV) are well-established approaches for estimating streamflow (FUJITA and KOMURA, 1994; Watanabe et al., 2021). Whilst operationalised systems using LSPIV exist (Bechle et al., 2012) using cross-correlation of sequential image vector fields, surface structure image velocimetry (SSIV) is a

derivative of LSPIV that filters the background enhancing the moving surface structures (Leitão et al., 2018). SSIV has been applied in the DischargeLab analysis software and DischargeKeeper operational monitoring system reaching technology readiness level (TRL) 9 in-use internationally with the ability to apply optical water level detection techniques with varying success without the use of a vertical gauging reference structure in the water (Photrack AG, Zürich, Switzerland) (Peña-Haro et al., 2021). Furthermore, STIV quantifies the change in luminance variation through time across one-dimensional search

lines defined parallel to the stream flow (Fujita et al., 2007). Whilst the majority of STIV trials have been part of research efforts, the approach has been packaged into user-friendly Hydro-STIV software (Hydro Technology Institute Co., Ltd.) and is being deployed by numerous organisations globally, particularly using unmanned aerial vehicles (UAVs) (Koutalakis et al., 2019). The central challenges with the application of these approaches remains the reliance on an externally measured water level, accurate ground control reference points assigned to fixed pixels in the frame for accurate vector transformation, and a

moderate degree of expertise required to manually tune site specific settings to reduce errors from changes in the site environment, and lighting conditions in initial setup (Detert, 2021).

With increasing successful developments in a range of methods in optical surface velocimetry, recent technological advancements in optical technologies for capturing videos, surveying environments, and computer vision analysis, along with technical advancements in embedded computing power efficiency, communications, and cloud computing/storage services,

we anticipate optical approaches to stream gauging will further transition from the research domain towards the operational domain.

The existing barriers to widespread implementation of optical stream gauging include; initial surveying and calibration of new sites, development of system integration, and difficulties in measuring velocities reliably with surface tracers across different site flow conditions, water clarity, and lighting environments (Pizarro et al., 2020). This study aims to develop and evaluate a

significant advance towards a rapidly deployable, accessible, automated operational, and scalable optical stream gauging system with improved reliability for gauging streams across varying flow and lighting conditions.

## 2 Methods

To address the study aim, the methodology firstly presents the operation of the computer vision stream gauging (CVSG) system and the process for site setup/configuration. Stereographic remote water level estimation approaches and adaptive cross section

learning, with rectification to coordinates of the water surface are then outlined. The optical flow technique used by CVSG for estimating surface velocities is then described along with the approach for learning the surface velocity distributions over



multiple measurements under different optical conditions, and fitting these surface velocity measurements to a model of the surface velocity profile transformed into a function of a boundary distance factor. Thereafter, the principles for the development of adaptive learning discharge ratings are detailed, followed by a summary of the reference operational discharge estimation approaches applied in this study. Finally, the characteristics of the optical flow field study sites, and the stream gauging field sites demonstrated in this work are presented.

## 2.1 Computer vision stream gauging system

The CVSG system employed in this study was developed for use in capturing stereo videos of waterways with automated processing of these into estimates of the water level, surface velocities, and gauged discharges. Figure 1 outlines the operational process of the CVSG system. The CVSG hardware has been designed around the use of a ZED 2/2i stereo camera (Stereolabs Inc., San Francisco, CA, USA) with or without internal infrared filters, and a NVIDIA Jetson Xavier NX (NVIDIA Co., Santa Clara, CA, USA). A sliding lens mechanism is inbuilt to allow for switching between different light wavelength band filters to enhance night measurements and collect data for discerning variables of water quality, such as suspended sediments. The system also employs a cloud architecture for automated data handling and internet of things (IoT) fleet management in managing the configuration of CVSG devices and sites drawing from a range of internal and external data sources. The integration of modern cloud analytics and fleet management allows for artificially intelligent predictive and adaptive sampling. Under typically configured baseflow conditions, the device operates at a sampling frequency of 60 minutes, capturing a video duration ranging between 3 and 30 seconds. During rapid rises in streamflow, the system can rapidly adapt and increase sampling frequency to between 2 and 15 minutes (depending on bandwidth and power conditions) to increase data density in less certain regions of the discharge rating.

Placing a CVSG device perpendicular or parallel with a waterway and configuring the upstream and downstream boundary distances relative to the camera that define the region to be analysed is sufficient for the system to begin estimating the level of any water within 40 m in-view of the camera. Then, using the inertial measurement unit (IMU), the accelerometer provides the orientation of the camera relative to gravity, and then projects the estimated planar water surface into the image space for rectified surface velocities in this plane to be estimated at 0.1 m resolution across the cross-section using an optical flow algorithm. Beyond this, providing the bathymetry of the cross-section and the camera's two-dimensional location (horizontal and vertical coordinates) relative to the reference frame of the cross-section data, allows the system to begin learning a cross-section model of the ground at a site, as well as fitting the model of surface velocity profiles, estimating discharge, and adaptively learning a discharge rating through hundreds and thousands of recordings at a site over time. Site configurations are stored and referenced in timeseries, allowing for site configuration changes through time, typically from updated manual cross-section surveys or changes to the camera location. It is possible to setup the system with a cross-section without knowledge of a surveyed location of the camera. Cross-section data can be referenced to the camera location visually through the projection of the cross-section data into the image overlay and tuning the predicted location of the camera to match the projection.





**Figure 1: Computer vision stream gauging (CVSG) system operation diagram.**

When videos are captured by the system, the analysis of these videos occurs in configured branches allowing the simultaneous automated analysis of the same video using different configurations or water level data sources (e.g. the original stereo-derived water level estimation, and an external water level sensor). Site coordinate systems are standardised with the x axis locally parallel to the waterway (positive in downstream direction), the y axis locally perpendicular to the waterway (positive away from or to the right of the camera), and the z axis aligned locally with the gravitational force measured by the camera. Cross-section data is one-dimensional and referenced to the y-axis across the waterway. The analysis region is not needed to be directly in front of the camera, but should be ideally a section of uniform and straight cross-section in view (can be as small as 0.5 m along the stream, up to as large as visible and practicable for optical flow measurement). When selecting a site, care



should be taken to identify sites with suitable surface flow visibility and oriented south-facing (southern hemisphere) or north-facing (northern hemisphere) where possible, while keeping the horizon above the line of sight of the camera. The CVSG system has been developed to balance ease-of-setup, ease-of-operation, affordability, accuracy of results, and reliability for stream gauging.

## 2.2 Stereo cross-section and water level estimation

The primary driver for the use of a stereo camera in the CVSG device is the potential to use stereophotogrammetry to reduce the surveying requirements typically associated with surface velocimetry techniques for the rectification of pixel displacements into realistic spatial scales over a wide range of water levels. However, a stereo computer vision system also makes it possible to initially survey and then continuously monitor the terrain of the cross-section above the water level for changes due to erosion, deposition, or vegetation. The adaptive learning of stream bank profiles over time allows for an advancement forward with non-stationary stream gauging in morphologically unstable sites.

With a point cloud calculated for each video frame, the median coordinate is taken of each of the three coordinate dimensions of the point cloud for each recording analysed. Water level is estimated by scanning across the point cloud in 0.5 m wide lines within the configured stream cross-section analysis region from the near bank towards the far bank. The median elevation between these cross-section scanning lines is taken, and the resulting cross-section profile is lastly filtered by a 0.5 m footprint median filter. After this, an iterative process constrains the near and far boundaries to ideally within the first 2 m (and not more than 10 m due to the effects of various optical conditions on the accuracy of the point cloud over the water surface) from the near bank across the water surface by moving the far boundary closer using similar pixels assumed to be water from the RGB image frame while avoiding any obstructed view of the near bank. The first percentile of the elevation points of the stereophotogrammetry cross-section profile within this domain is then estimated as the water level.

The CVSG system has also been developed with external water level data source aggregation and parallel analysis capability alongside stereo water level estimates. With an estimated water plane level relative to the camera position, and an IMU (median filtered and stability tested for each recording) providing the orientation of the camera relative to gravity, as well as the camera's optical properties, the across stream coordinates of the water surface plane every 0.1 m are first projected and interpolated into the image pixel space. Following this, the streamwise coordinates along the stream are predicted for each pixel spanning from the image centreline.

## 2.3 Optical flow surface velocimetry estimation

The motions in the recorded videos are computed using the Farneback algorithm (Farnebäck, 2003) to solve the energy-like minimisation problem for the optical flow equation across the pixel space between each pair of consecutively recorded frames. Whilst optical flow has previously been applied to the measurement of stream discharge (Perks, 2020; Khalid et al., 2019), there are different existing algorithms developed for reaching an optimal solution of the optical flow equation (Baker et al., 2011). Shi et al. (2020) compared three established and widely applied optical flow techniques to breaking surges, noting the




advantages of the Farneback algorithm for its relatively high accuracy and dense flow fields, as well as a lower sensitivity to noise. The approach combines the assumptions of local neighbourhood brightness intensity variation between frames with minimised energy and the global minimisation of an energy function assuming a slowly varying displacement field for locally

smooth velocity gradients (Shah and Xuezhi, 2021). A four-level pyramid of processing steps (Adelson et al., 1984) is applied to estimate larger overall motions first at a coarser resolution, interpolating these larger motions to higher resolutions over the four steps refining the optical flow field with each step increasing in resolution up to the original video resolution (typically 1920x1080 recorded from each camera simultaneously).

With the optical flow algorithm applied to each pair of consecutively recorded frames, depending on the visual flow conditions

there can be errors in the estimated motion between frames, camera vibrations, as well as natural or man-made motions occurring between the camera and the waterway surface to be measured. By summing the estimated optical flow fields over the duration of each recording, and averaging the optical flow field stack produced to find the nett motions estimated over the duration of each recording, any oscillatory and non-continuous motions can be supressed. Taking the two-dimensional image space gradients of the previously computed streamwise (x) and cross-stream (y) coordinates of the water surface plane, the

optical flow pixel displacement rates are scaled onto the water surface plane, noting that the optical flow motions in the horizontal and vertical directions of the image space can each be indirectly measuring components of both the streamwise and cross-stream motions on the water surface plane. From this point, the motions out of the plane of the water surface are filtered out of the analysis to further remove false motions unrelated to the waterway surface velocities.

Assuming the remaining velocities over the length of the analysis section are velocities related to the motion of the water

surface, a continuity of streamlines is applied with the strongest detected velocities collapsed into a single-dimensional raw cross-section surface velocity profile. The assumed continuity of streamlines within the analysis section length facilitates the measurement of velocities across spatially inconsistent optical flow measurement/lighting conditions along the length of the analysed section.

Whilst the visual conditions for optical flow measurement can still be insufficient for reliable measurements across the length

of the measurement section in all situations, the CVSG system then applies an adaptive learning velocity distribution across the cross-section at 0.1 m intervals and references these learning measurements over the observed water level range at 0.01 m intervals. This process of developing an adaptive database of surface velocity measurements across the stream at different water levels, allows the system to use multiple measurements of the same water level over time in different conditions to combine these measurements into a complete velocity profile, while simultaneously being adaptive to observed changes in

surface velocity profiles in non-stationary environments. Furthermore, there can still be biases present in these measurements over time which could take the form of incorrect velocity signals and sections of the cross-section which are persistently in poor optical flow measurement conditions or entirely out of the range of the camera's pixel resolution in order to measure the displacements with any accuracy. In this case data gaps are filled by fitting the sufficiently measurable surface velocities to an exponential relationship model in a transformed spatial domain that scales with each measured surface velocity's relative

distance to the boundaries of the flow according to Eq. (1):



$$v_s = v_\infty\left(1 - e^{-bx}\right),\tag{1}$$

where $v_s$ is the surface velocity, $v_\infty$ constrains the asymptote approaching the free stream velocity, $b$ is a cross-section variable, and $x$ is the boundary distance factor (defined here as the depth multiplied by the distance to the nearest water surface edge). The relationship in Eq. (1) bounds the surface velocity to zero where either the depth reaches zero or at the intersection

of the water surface and the cross-section. By using the measured adaptive learning surface velocities distribution as data for automated fitting of the gap-filled surface velocity profile model across the entirety of the cross-section, an estimated surface velocity profile consistent with the measurements collected at each water level is produced. The Trust Region Reflective algorithm (Branch et al., 1999) is used to optimise the least squares fit of this data by predicting the free stream velocity and cross-section variables fitting the measurements learned in the adaptive surface velocity distribution. Whilst the automated

fitting of the surface velocity profile model assumes that the relationship between the measured surface velocities and the flow boundary are consistent across the cross-section, this assumption is likely to weaken with significant variations in channel roughness across the cross-section, particularly if the surface velocities neighbouring these regions of different roughness are not represented by samples in the adaptive surface velocity distribution.

## 2.4 Adaptive learning discharge rating

In order to estimate discharges from surface velocity profiles, an assumption is made in scaling the surface velocities to approximate the mean velocity profile across the cross-section using a ratio, $\alpha$, which is then integrated over the cross-section area at 0.1 m increments to estimate discharge. Hauet et al. (2018) examined the vertical profiles of 3611 gaugings from 176 sites with different bed types (concrete, sandy, pebbly, boulders), finding the primary driver of the ratio, $\alpha$, to be the depth of flow. Their study found a linear trend was found for hydraulic radiuses above 1 m ($\alpha = 0.8$) up to 5 m ($\alpha = 0.9$), and

furthermore it was concluded to use $\alpha = 0.8$ for depths less than 2 m, and $\alpha = 0.9$ for depths greater than 2 m in natural channels. Following from this, the CVSG analysis applies a varying $\alpha$ across the cross-section switching between a low (default 0.8) and high (default 0.9) value dependent on a threshold depth (default 2 m) at each 0.1 m interval. The result of this cross-section varying depth-dependent approach is an effective $\alpha$ weighted on the distribution of depths within the cross-section.

Using the adaptive learning surface velocity distributions and the associated fitted model surface velocity profiles, an adaptive discharge estimation is produced for each observed water level at 0.01 m intervals. This process is replicated independently in parallel for the lower and upper surface velocity and discharge estimates to produce an adaptive learning discharge rating envelope. The result of this is a new discharge rating envelope fitted to the latest discharge estimates across all of the observed water levels at a site with each new measurement. The learning discharge rating can be configured to be filtered with a locally

fitted Savitzky-Golay signal filter (Savitzky and Golay, 1964) (a filter window size of 0.05 m vertical with nearest boundaries and linear fitting is used for the results presented in this work) and power law weighted by the number of observations and optical flow coverage measured at each 0.01 m water level increment. Quality codes are automatically determined for each




discharge estimate based on a function of the number of observations, the optical measurement coverage resulting from the lighting and seeding conditions, water level estimation, and convergence between the current measurement, the learning velocity distribution, the fitted surface velocity profile model, and the learning discharge rating.

**2.5 Operational discharge estimation references**

Results from the CVSG derived discharge estimates in this study have been directly compared against the best available acoustic and optical methods for measuring discharge. As part of these reference technologies, two commercially available and well-developed technologies for estimating discharge using optical methods of measuring surface velocities have been applied. These analyses were undertaken by DischargeLab software (Photrack AG, Zürich, Switzerland) using surface structure image velocimetry (SSIV) (Leitão et al., 2018), and Hydro-STIV software (Hydro Technology Institute, Osaka, Japan) using space-time image velocimetry (STIV) (Fujita et al., 2007). Furthermore, the raw surface velocity results from these technologies has been processed using the surface velocity model fitting methodology used in the CVSG system that is presented here demonstrating the broader applicability of the methods presented in this study.

Acoustic Doppler Current Profilers (ADCPs) were utilised in order to estimate the subsurface velocities and produce reference acoustic estimates of the discharge independent of the optical approaches. Where available, ADCP velocity estimates closest to the surface (between 0.13 to 0.19 m depth) were compared to surface velocity estimates from the optical surface velocimetry technologies. Additionally, historical ADCP derived estimates of discharge used to develop discharge ratings were utilised as a reference, along with the most up to date published discharge rating fits available. These latest discharge rating fits at water level gauging station sites were then also used as a reference for timeseries comparisons through the conversion of the recorded water levels to the discharge predicted by the discharge rating fit. Although these different reference estimates have their own limitations and uncertainties, the comparison of the best available estimates at each case study site using the different technologies can provide some insights both in where they differ and to what degree they agree. At two existing government maintained gauging stations, historical manual gaugings have been compared along with CVSG, DischargeLab, and Hydro-STIV measurements relative to the latest published discharge rating using root-mean-square error (RMSE), the mean percentage difference, and the Nash-Sutcliffe Efficiency (NSE) (Jackson et al., 2019) commonly applied for assessing predictive skill for discharges in hydrological settings.

**2.6 Field case study sites**

This study includes four field case study sites (Figure 2), inclusive of a single surface velocimetry benchmark time captured on the Castor River, Ontario, Canada (Perks et al., 2020), and a single capture of an irrigation channel in NSW, Australia. Single points in time were focused on the optical method assessments relative to the available reference data for measuring surface velocities and estimating discharges. Two of the four sites presented here were continuously gauged with CVSG devices on the Tyenna River, Tasmania, Australia, and on the Paterson River, NSW, Australia. One point in time recorded from Tyenna River, Tasmania, Australia in the middle of the observed water level range was also used for assessment of the



optical methods relative to the available reference data. In total, 18 recorded points in time spanning the observed water level
range were evaluated between the discharge estimates of the optical methods at Tyenna River, Tasmania, Australia.
Furthermore, both the Tyenna River and Paterson River CVSG deployments were evaluated with long-term operational
considerations for the implementation of CVSG methods into routine stream gauging. Table 1 provides a summary of the
available reference data for the field case study sites presented in this work, as well as the measurement ranges observed.

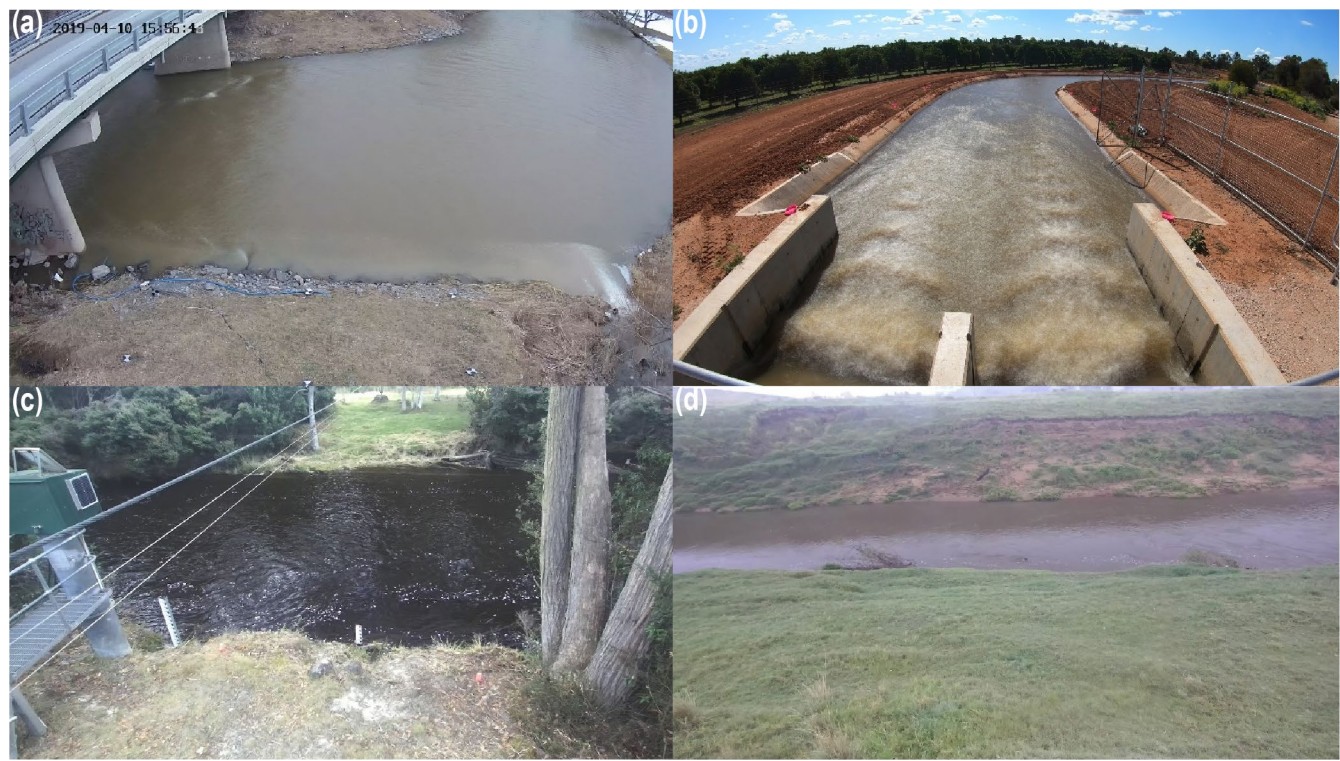

**Figure 2: Images collected from study sites at (a) Castor River, Ontario, Canada [10 April 2019 15:55 LT], (b) an irrigation channel, NSW, Australia [19 September 2020 14:00 LT], (c) Tyenna River, Tasmania, Australia [5 March 2021 12:12 LT], and (d) Paterson River, NSW, Australia [1 March 2022 09:48 LT].**

**Table 1: Field case study sites summary.**

| Site | Period | Distance to stream (m) | Water level (m) | Reference gaugings | Ground control reference points |
|---|---|---|---|---|---|
| Castor River, Ontario, Canada | 30 s | - | 3.77 | 1 concurrent (2019) | 12 |
| Irrigation channel, NSW, Australia | 30 s | - | 135.80 | 1 concurrent (2020) | 10 |
| Tyenna River, Tasmania, Australia | 56 d | 5.9–7.3 | 0.31–0.87 | 344 historical ('64 – '22) | 9 |
| Paterson River, NSW, Australia | 122 d | 0-22.5 | 0.78–10.54 | 157 historical ('87 – '21) | 0 |



### 2.6.1 Castor River, Ontario, Canada

The surface velocity profiles and discharge estimates for the Castor River case study site at Russell in Ontario, Canada orientated facing across the stream from the left bank were analysed in this work utilising the published benchmark data (Perks et al., 2020) from 10 April 2019 where an approximately 20 Hz 30-second duration video recording was captured at 15:55 local time (LT) (2688x1520 pixel resolution). The benchmark data included a reference Teledyne RDI StreamPro ADCP (Thousand Oaks, CA, USA) moving boat transect with 0.05 m vertical cell resolution and the topmost cell measuring at 0.17 m depth. From the cross-section depth characteristics, the surface velocities extrapolated from the ADCP transect were scaled using the same 0.8 ratio assumed to approximate the depth-averaged velocity in the surface velocimetry approaches, and then integrated across the channel cross-section to estimate a comparable reference discharge. However, the most recently published gauging station rating was ultimately used as the reference discharge estimation, as this is currently the reported discharge that is estimated when the water level (3.77 m) is measured by the gauging station. This benchmark case study presents a softly lit environment with visible surface rippling features across the full width of the cross-section, and a sky/vegetation reflective water surface. There are twelve ground control reference points provided with the recording for spatial rectification and scaling of pixel displacements over time. The cross-section recorded by the moving boat ADCP transect was used for the estimation of discharge and boundary distance factor surface velocity profile model fitting with an approximate maximum depth of 1.2 m over a 27 m wide cross-section (average depth 0.8 m). Surface velocity analysis regions for all technologies utilised were conducted in a similar region across the downstream side of the ground control reference points closest to the bridge to the left of frame. To consider the variation resulting from different recording durations, raw CVSG surface velocities were analysed over the cross-section for recording durations of 5 seconds, 10 seconds, and 20 seconds.

### 2.6.2 Irrigation channel, NSW, Australia

Canal channels are an important waterway type globally for the measurement of discharge, with countries such as the United Kingdom containing over 600 000 km of channels/ditches, while streams/rivers comprise some 270 000 km (Peacock et al., 2021). A field case study was undertaken with a camera oriented facing downstream from a hydraulic sluice gate control structure in an irrigation channel in NSW, Australia on 10 September 2020 at 14:00 LT. A 30 Hz 30-second video recording (3840x2160 pixel resolution) formed the basis for the surface velocimetry estimations, with a reference measurement provided by SonTek RS5 moving boat ADCP (San Diego, CA, USA) transects taken between 15 to 20 m downstream of the hydraulic control structure. A non-uniform flow distribution is evident across the width and length of the irrigation channel with evidence of standing-waves at the free surface including shimmering sun glare reflections. The area of the irrigation channel was surveyed with ten ground control reference points for rectification of the image spatial scales and surface velocities from the pixel displacements over time. The cross-section used for discharge estimation was taken from the ADCP reference transects with a maximum depth of 1.3 m and a transect width of 6.8 m. CVSG optical flow analysis along the length of the channel showed a marked reduction in flow visibility towards the region of the reference ADCP transects downstream (Figure S1).



Furthermore, the optical surface velocity measurement technologies were optimised to their suitable regions of interest. Hydro-STIV required a sub-optimal analysis region closer to the camera and hydraulic structure in order to estimate 1-dimensional space-time image stack angles for the determination of velocities, necessitating search line lengths which were unsuitably short for analysis under these conditions. Additionally for this case study, the CVSG analysis region was limited to 1 m channel section lengths as the assumptions for CVSG analysis were found to be violated for sufficiently large analysis regions along the length of the cross-section where the streamlines were not continuous. Beyond this, the CVSG raw surface velocities using

recording durations of 5 seconds, 10 seconds, and 20 seconds have been analysed to assess the difference in measurements over differing sampling durations.

### 2.6.3 Tyenna River, Tasmania, Australia

The first long-term CVSG device field site was installed at the Tasmanian Government gauging station site at Newbury on Tyenna River, Tasmania, Australia with CVSG analysis beginning from 5 March 2021 at 16:47 LT. A 216 km$^2$ catchment

upstream provides continuous flow through the site with observed water levels between 0.26 and 0.87 m (maximum depths between 0.68 m and 1.29 m) during CVSG operation. Along with a cross-section of the site's bathymetry, nine ground control reference points were surveyed in the field of view of the camera in order to carry out reference analyses with DischargeLab and Hydro-STIV, while none were used for the CVSG analysis. The bed of the stream becomes visible at low water levels (< 0.4 m) with very little visibility of the water surface. Whilst the site experiences significant variations in lighting and flow

conditions, the site's streamflow conditions are considered well-suited for remote optical measurement of surface velocities due to naturally occurring coverage of surface features above baseflow. However, it should be noted that the onset of standing waves was evident at the upper end of the range recorded by the CVSG device.

The CVSG device was mounted at an approximate water level of 6.32 m at a distance from the near bank water edge which ranged between 5.9 to 7.3 m over the recorded water level range. Video recordings were generally set to be taken with durations

ranging between 5 to 20 seconds at 10-minute intervals generally with a resolution of 1920x1080. The system was offline for 5 months between May and October 2021 during the first 12 months of operation included in this study. The CVSG analysis was configured to consider a constant 9 m long analysis region along the length of the stream with the positive flow direction provided to the right of the camera. A standard point cloud average temporal variation tolerance of 0.1 m was also set.

Historical manual discharge gauging measurements were provided for the site since 1964 averaging between 2 to 9 gaugings

340    per year each decade over a water level range from 0.23 to 1.54 m with the peak number of gaugings in the 1980s. There were 18 reference comparison time points selected from the CVSG recordings in March and April 2021 covering the range of water levels with a variety of lighting conditions. One of these comparison time points (30 March 2021 12:12 LT at a water level of 0.509 m) was used for the more detailed comparison of the estimated surface velocity profiles between the technologies. The most recent manual gauging measurement recorded within 0.05 m of the time point of the more detailed comparison time

point's water level was also used for reference. This manual gauging measurement was undertaken on 7 August 2019 at 11:32



LT using a SonTek M9 RiverSurveyor moving boat ADCP setup (San Diego, CA, USA) on the fixed cableway existing at the site.

### 2.6.4 Paterson River, New South Wales, Australia

A second long-term CVSG device field site was installed at the WaterNSW gauging station at Gostwyck on Paterson River, with analysis beginning from 3 November 2021 05:59 LT and finishing after being submerged on 4 March 2022. Significant flows with stream rises greater than 10 m that are ephemeral in nature at this site are more representative of regional Australian rivers. With a catchment area of 956 $km^2$ which is largely cleared for agricultural land use, 2 km upstream from the site is a confluence with the Allyn River, and Lostock Dam is 80 km upstream. Over the 4 months of CVSG operation at this site, water levels between 0.78 and 10.54 m were recorded (maximum depths ranging between 1 and 11 m) with cascading erosion of the far bank captured by the CVSG device at water levels higher than baseflow. Strong winds along the streamflow direction were observed in the first month of deployment significantly biasing low flow surface velocities.

The CVSG device was installed perpendicular to the streamflow at an approximate water level of 11.34 m at a distance from the near bank water edge which ranged between 0 and 22.4 m over the recorded water level range. A cross-section of the bathymetry was surveyed in a straight line along the same streamflow perpendicular axis as the camera orientation up to a water level of 8.38 m (2.2 m below the highest observed water level during CVSG operation), noting the location of the camera along this cross-section survey line. No ground control reference points were surveyed at this site, and there was therefore no comparison to other optical surface velocimetry technologies available. While alternative measurements for direct comparisons were not available, the latest published discharge rating estimates as well as the 157 historical manual gauging measurements taken across water levels ranging from 0.56 to 10.54 m since 1987 were used for reference. Dynamically varying downstream and upstream analysis boundaries were defined entirely to the right hand (upstream) side of the CVSG device. Below 2 m water level, the analysis region was set to an 18 m channel length, while water levels above 2 m were able to analyse an expanded 35 m long channel length. The video recordings taken at the site were 10-second duration at a resolution of 1920x1080 every 10 minutes.

### 3 Results

The results of this work found broadly comparable gauging results using the raw data of the different measurement technology approaches employed, predominantly falling within a relative error of 15% under suitable conditions. However, it was demonstrated that the use of a surface velocity profile model fitted to raw measurements under suboptimal or only partially measurable conditions could be beneficial to improving the reliability of surface velocimetry methods. Furthermore, the learning of an adaptive surface velocity distribution extended to an adaptive learning discharge rating produced robust results for stream gauging over time. While the initial CVSG results for the non-contact measurement of water levels using




stereophotogrammetry found less than 20% of measurements within 0.05 m, to the learning capability of CVSG was able to converge towards a robust discharge rating despite noisy raw observation data.

## 3.1 Optical surface velocimetry

The surface velocity and ADCP (0.17 m depth) profiles across the cross-section distance for the Castor River case study (Figure 3a) measured velocities increasing from the bank up to 1.5 m/s occurring by approximately 2 m across the channel for all measurement technology approaches. Within the mid-section of the channel, velocities were measured between 1.5 and 2 m/s, with Hydro-STIV observing the lowest peak velocity, less than the peak measurement of the ADCP beneath the surface (which recorded some velocities on the order of 2 m/s). The lower resolution of Hydro-STIV search line measurements across the channel could be attributed to this result missing the peak velocities measured by the other technologies, however the other reference technologies consistently estimated higher velocities across the entire channel mid-section. On the other hand, Hydro-STIV showed the highest rate of full velocity development from the edge of the channel, with the raw CVSG surface velocities estimating the lowest rate of full velocity development from the edge of the channel. Furthermore, the raw DischargeLab and CVSG velocity estimates tracked closely across the channel, although the optically estimated surface velocities were all in agreement with peak surface velocities occurring approximately 2 m closer to the left bank of the channel relative to the peak velocities measured by the ADCP. It can be seen that the optical surface velocity measurement technologies were able to measure velocities in the shallower regions closer to the channel edges than was possible with ADCP moving boat transects.





**Figure 3: Detailed time point comparison raw and model fitted velocity measurements plotted over the cross-sections at (a) Castor River, Ontario, Canada, (b) an irrigation channel in NSW, Australia, and (c) Tyenna River, Tasmania, Australia. (d) Correlation plot between the gauge rating and optically estimated discharges at comparison time points at Tyenna River, Tasmania, Australia, with the detailed comparison time point indicated. CVSG 5-second duration surface velocities shown for (a) Castor River, Ontario, and (b) the irrigation channel in NSW, Australia. CVSG 10-second duration surface velocities shown for (c, d) Tyenna River, Tasmania, Australia.**

With the application of the CVSG surface velocity profile model fitting methodology to all three optical surface velocity estimation approaches (Figure S2a), it can be seen that whilst the raw surface velocity estimates largely fit within the bounds of the ADCP measurements at 0.17 m depth over the scale of the boundary distance factor, the resulting surface velocity profile model fits from each technology takes on a different shape. However, despite these differences in fitted surface velocity model profiles with the free stream surface velocities fitted ranging from 1.55 m/s in the case of Hydro-STIV, up to 1.9 m/s in the case of CVSG, as well as differences in the cross-section variable ranging between 0.45 (for CVSG) and 1.7 (for Hydro-STIV), the resulting discharges between all reference comparisons (including varying recording duration for raw CVSG analysis





between 5 and 20 seconds, and the ADCP discharge estimation using constant extrapolation to surface velocity) were within 5% of the latest published gauging station rating discharge at the recorded water level (Table 2). For this case study, there was little sensitivity observed between the different measurement technologies, the duration of recording, or the resulting fitted
surface velocity profile model parameters on the estimated discharges. Overall, the surface velocity profile model fitting methodology did not negatively impact the calculation of discharge for any of the surface velocity estimation technologies in this case study.

**Table 2: Summary of discharge estimates for gauging technologies at Castor River, Ontario, Canada (10 April 2019 15:55 LT).**

| Measurement | Type | Duration | Discharge (m³/s) |
|---|---|---|---|
| **Water Level** | Published rating @ 3.77 m | - | 27.8 |
| **ADCP (surface)** | Constant extrapolation to surface | N/A | (+1%) 28.2 |
| **CVSG** | Raw | 5 s | (-2%) 27.3 |
| | Raw | 10 s | (-4%) 26.8 |
| | Raw | 20 s | (-4%) 26.6 |
| | Model fit [$v_\infty = 1.9$, $b = 0.45$] | 5 s | (+1%) 28.1 |
| **DischargeLab** | Raw | 25 s | (+4%) 28.9 |
| | Model fit [$v_\infty = 1.8$, $b = 0.8$] | 25 s | (+3%) 28.6 |
| **Hydro-STIV** | Raw | 25 s | (+1%) 28.0 |
| | Model fit [$v_\infty = 1.55$, $b = 1.7$] | 25 s | (+2%) 28.3 |

In the irrigation channel case study, the raw surface velocity profiles estimated were broadly in agreement about the shape of the velocity profiles across the channel (Figure 3b), with the largest peak occurring within 2 m of the left bank, as well as reduced surface velocities measured in the middle of the channel and increased velocities towards the right bank. These velocity profiles are consistent evidence of the influence of the two hydraulic control gates upstream releasing water into the channel.
Whilst the surface velocity magnitudes measured by the raw CVSG analyses from durations of 5, 10, and 20 seconds across the channel profile were more consistent with the ADCP measured velocities at 0.13 to 0.19 m depth, the surface velocities estimated by Hydro-STIV and DischargeLab were on the order of double within 2 m of the channel edges. While the objective truth of the surface velocity profile across the channel is not known, the Hydro-STIV analysis contain ambiguous results with unclear space-time pattern angle identification under these conditions. Furthermore, this case study not only demonstrates
where site selection can lead to ambiguous optical surface velocity measurements, but also highlights a case where the assumptions of the CVSG surface velocity profile model are not valid, and therefore should not be applied (as in Figure S2b). Although the discharge is only inferred from the signals measured by the reference technologies, only the raw CVSG measurements and ADCP discharge estimates agree within 5% across all recording durations analysed (Table 3).





With 133 samples measured over the length of the ADCP cross-section, the resulting ADCP measurement resolution was on
the order of 0.04 m per sample (with 1 second per sample). This meant that the ADCP sampling durations for each segment of
the channel was on the order of 2.5 seconds per 0.1 m channel cross-section segment. The lowest duration CVSG optical
surface velocity analysis occurred over 5 seconds sampling the entire cross-section simultaneously at 0.1 m resolution (not
possible with a moving boat ADCP transect) which resulted in a 1% to 4% difference to the ADCP discharge estimate.
However, the incorrect application of the surface velocity profile model resulted in estimates 55% higher than the reference
ADCP discharge estimate.

**Table 3: Summary of discharge estimates for gauging technologies at an irrigation channel in NSW, Australia (10 September 2020 14:00 LT).**

| Measurement | Type | Duration | Discharge (m³/s) |
|---|---|---|---|
| **ADCP (profile)** | Moving boat | 124 s | 1.65 |
| **CVSG** | Raw | 5 s | (+2%) 1.68 |
| | Raw | 10 s | (-1%) 1.64 |
| | Raw | 20 s | (-4%) 1.58 |
| | Model fit [$v_\infty = 0.8, b = 0.4$] | 5 s | (+55%) 2.56 |
| **DischargeLab** | Raw | 20 s | (+215%) 5.19 |
| | Model fit [$v_\infty = 1.05, b = 2.5$] | 20 s | (+224%) 5.34 |
| **Hydro-STIV** | Raw | 20 s | (+111%) 3.48 |
| | Model fit [$v_\infty = 1, b = 1$] | 20 s | (+189%) 4.77 |

After 25 days into the operation of CVSG at the case study site on the Tyenna River capturing 2452 gaugings, the first flow
event had been observed (reaching a maximum recorded water level of 0.735 m) with the water level receding to 0.509 m on
30 March 2021 at 12:12 LT, the comparison surface velocity profile estimates were found to be grouped within ± 0.1 m/s in
the mid-section of the channel (Figure 3c). The ADCP measured mean velocities across the channel were scaled up to estimate
the surface velocity using the assumed ratio of 0.8 recorded at a water level 0.038 m higher on 7 August 2019 at 11:32 LT.
Similarly, this resulted in ADCP estimated surface velocities within the same range of variability measured by the surface
velocimetry reference methods, but with a 14% higher calculated discharge relative to the latest published discharge rating at
0.509 m water level (Table 4). Although the true discharge is not known, all reference technologies estimated discharges within
10% of the latest published discharge rating at this water level, except for the ADCP profile recorded 601 days prior at a higher
water level estimating within 15%.
Surface velocity profile model fits for this case study measurement ranged between 0.78 (CVSG and Hydro-STIV) to 0.88 m/s
(DischargeLab) for free stream velocities, and 1 (DischargeLab) to 1.4 (Hydro-STIV) for cross-section variables (Figure S2c).
The model fitting of the raw surface velocities estimated across all optical measurement technologies was not found to have





any negative measurable impacts on the calculation of discharge. However, in practice the model fit would not be computed
using the raw velocities alone, as the CVSG system would instead compute the model fit of the learning velocity distribution
after the 25 days of measurements that had resulted in 8 observations within 0.005 m and an accumulated 72% optical flow
measurement coverage over the width of the cross-section at this water level (Figure S3a).

Furthermore, up to this measurement time there had been a total of 2452 CVSG gaugings across a wider range of water levels,
and the resulting learning discharge rating was also estimated (Table 4). Additionally, quantifying any further change in the
learning estimations considering a point in time which is another 30 days advanced into the CVSG gaugings (Figure S3b) saw
the difference between the learning estimated CVSG discharges and the latest published discharge rating at 0.509 m water
level drop from within 6% to within 2%. However, it is important to note that these relative differences are expected to be
within the realm of uncertainty of the true discharge, particularly as the discharge has only been measured at this water level
once in 1966, with measurements within 0.005 m occurring five times (most recently in 1989), and 37 measurements within
0.05 m (the two most recent occurring 2 years and 8 years prior to the time of this case study recording) (Figure 4).


**Table 4: Summary of discharge estimates for gauging technologies at Tyenna River, Tasmania, Australia (30 March 2021 12:12 LT) using gauge water levels.**

| Measurement | Type | Duration | Discharge (m³/s) |
|---|---|---|---|
| **Water Level** | Published rating @ 0.509 m | - | 4.37 |
| **ADCP (profile)** | Moving boat 7 August 2019 11:32 LT @ 0.547 m | 1522 s | (+14%) 4.97 |
| **CVSG** | Raw | 10 s | (-6%) 4.09 |
| | Model fit [$v_\infty = 0.78$, $b = 1.2$] | 10 s | (-6%) 4.11 |
| *25 days gauged* | Learning model fit [$v_\infty = 0.86$, $b = 1.39$] - 8 observations @ 72% coverage | 10 s | (+6%) 4.63 |
| | Learning rating (2452 observations) | - | (+5%) 4.58 |
| *55 days gauged* | Learning model fit [$v_\infty = 0.8$, $b = 1.47$] - 19 observations @ 84% coverage | 5 s | (-1%) 4.33 |
| | Learning rating (3890 observations) | - | (+2%) 4.46 |
| **DischargeLab** | Raw | 10 s | (+5%) 4.59 |
| | Model fit [$v_\infty = 0.88$, $b = 1$] | 10 s | (+4%) 4.55 |
| **Hydro-STIV** | Raw | 10 s | (-8%) 4.03 |
| | Model fit [$v_\infty = 0.78$, $b = 1.4$] | 10 s | (-3%) 4.23 |





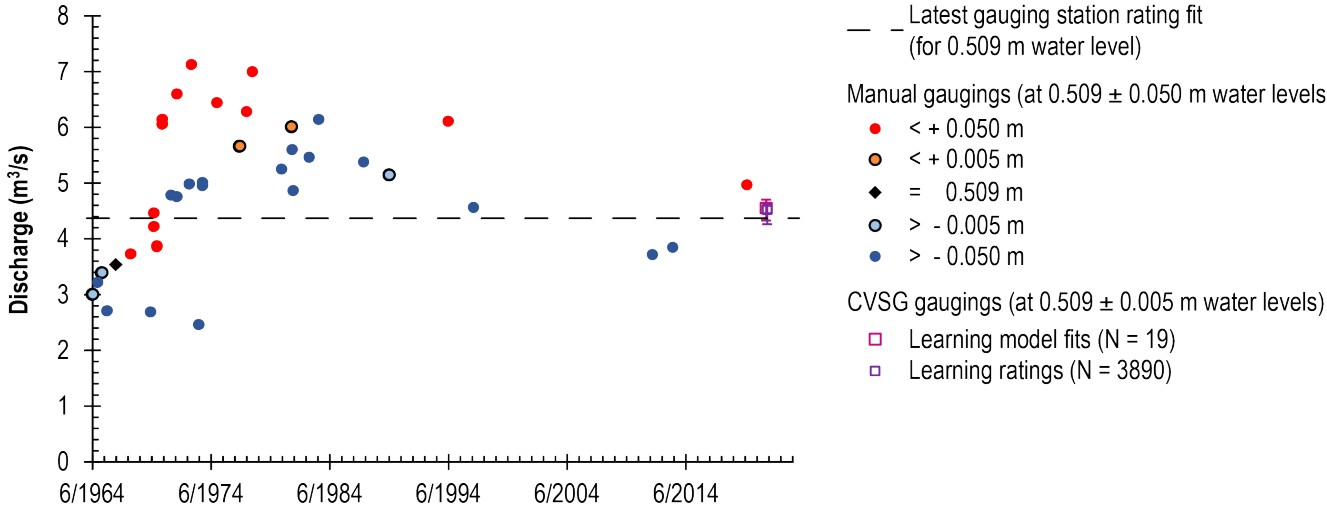

**Figure 4: Historical and CVSG gaugings at Tyenna River, Tasmania, Australia in the vicinity of 0.509 m water level.**

Further comparison of the reference optical surface velocimetry technologies at 16 time points along the timeseries recorded at Tyenna River show the importance of discharge gauging technology assessments over multiple conditions (Table 5). With the evaluation of more time points, an understanding of the most suitable conditions for gauging can be built from the statistics of the RMSE, mean percentage bias difference, and the NSE relative to the reference latest gauging station rating. In the absence of gauged water levels, the raw CVSG discharge estimation had the greatest timeseries absolute error performance with an RMSE of 1.28 m³/s, however suffered a worse bias in the mean difference relative to the latest gauging station discharge rating, with the learning estimations demonstrating improving this bias to 1.9% in hindsight after 12 months of learning with an increased 2.28 m³/s RMSE. For the CVSG discharge estimations using the gauge water level, the RMSEs were reduced to less than half of the stereophotogrammetry estimated water level-based discharge estimations, implying that the stereophotogrammetry estimation of water level is the largest source of error in the discharge estimate.






**Table 5: Summary of comparison time points relative to the gauge rating at Tyenna River, Tasmania, Australia.**

| Measurement | Type | RMSE (m³/s) | NSE |
|---|---|---|---|
| CVSG (stereophotogrammetry estimated water level) | Raw | 1.28 | 0.905 |
| | Learning model fit | 2.69 | 0.581 |
| | Learning rating | 2.26 | 0.705 |
| | Learning rating 12-month | 2.28 | 0.719 |
| CVSG (gauge water level) | Raw | 0.48 | 0.986 |
| | Learning model fit | 1.18 | 0.919 |
| | Learning rating | 1.03 | 0.939 |
| | Learning rating 12-month | 0.60 | 0.979 |
| DischargeLab (gauge water level) | Raw | 0.97 | 0.945 |
| Hydro-STIV (gauge water level) | Raw | 0.68 | 0.973 |


## 3.2 Stereophotogrammetry water level detection

During the operation of the CVSG system on the Tyenna and Paterson Rivers, water levels were remotely estimated using stereophotogrammetry with every recording possible, requiring sufficient daylight, as night vision equipment was not installed at these sites during the period of this evaluation. The timeseries of remotely estimated water levels were classified into

different error ranges relative to the water levels recorded by the reference gauging stations at Tyenna River (Figure S4a), and Paterson River (Figure S4b). Water level estimation errors within 0.005 m were considered to be exactly correct relative to the 0.01 m water level database precision used in CVSG operation in this work. Errors found to be within 0.05 m were also expected to be useful as this is the water level Savitzky-Golay filter window size used in the CVSG learning distributions of these studies. Both sites in this study with CVSG deployed suffered from higher variability during lower flows, corresponding

to low visual distinction of the texture of the water surface, particularly with the greater distance of the site to the edge of the Paterson River at low flows, and clear water displaying the bed of the Tyenna River.

Correlation plots between the gauge water level and the CVSG stereophotogrammetry estimated water level at Tyenna River (Figure 5a) and Paterson River (Figure 5b) show the remotely estimated water levels compared over the range of the gauge recorded water levels with some potential clustering of different error regions at particular water levels. From this, the

cumulative error distributions (Figure 5c) shows 2% and 1% of measurements taken over the study period were considered to be precise at the Tyenna River and Paterson River sites respectively. Furthermore, just 16% and 7% of measurements at the sites were considered within the CVSG learning distribution noise tolerance, with the remaining majority of measurements falling outside of this error range with potentially significant contributions to discharge bias possible as a result.



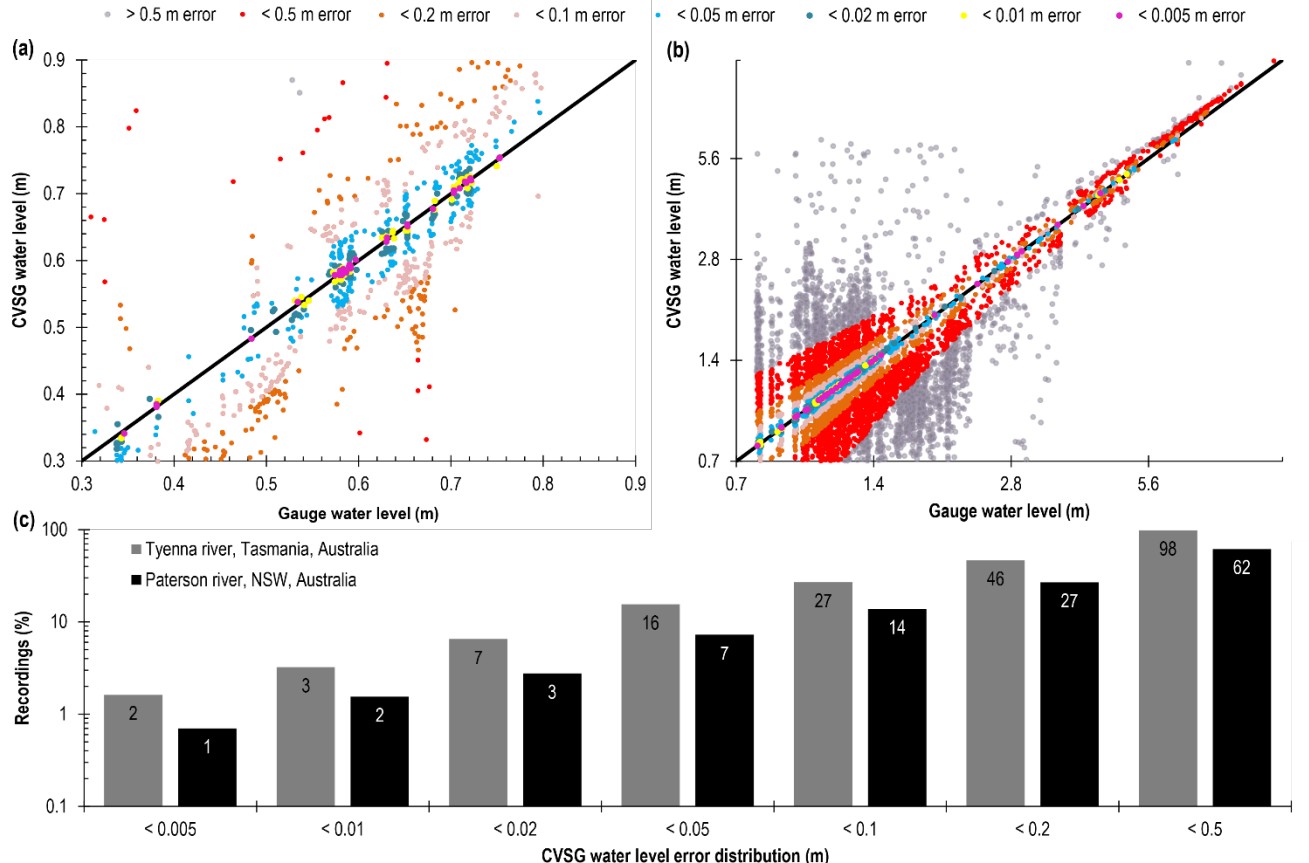

**Figure 5: Correlation between gauge and stereophotogrammetry estimated CVSG water level classified according to error magnitude at (a) Tyenna River, Tasmania, Australia, and (b) Paterson River, NSW, Australia on a logarithmic scale. The cumulative stereophotogrammetry estimated CVSG water level error class distribution (c) for both Tyenna River, Tasmania, Australia, and Paterson River, NSW, Australia on a logarithmic scale.**

### 3.3 Timeseries discharge

The real-time estimation of discharge over time relative to the latest gauging station rating at Tyenna River using CVSG raw, learning model fit, and learning rating, all demonstrated the ability to capture the patterns of hydrographic rises and falls despite the significant presence of errors in the remote stereophotogrammetry estimation of water level (Figure 6a/Figure S5a). Whilst the raw CVSG discharge estimates follow the latest gauging station rating estimation more closely than the learning estimations, this can be attributed to the significantly higher vulnerability of the learning estimates for any individual time

point to errors in the water level estimation. However, with the learning process continuing over 12 months at the site, a marked improvement in the discharge estimation from the learned discharge rating (despite the underlying water level estimation errors) was evident.

Using the gauge water level in the CVSG analysis independently parallel to the stereophotogrammetry estimated measurements yielded learning discharge estimations rapidly improving on the variability observed in the raw CVSG measurements (which



are each independently estimated through time) relying on the suitability of the naturally occurring conditions for optical flow
at the time of measurement (Figure 6b/Figure S5b). Even though the true discharge at the measurement times are not known,
the CVSG learning discharge estimations using the gauge water levels at the time somewhat overestimated the discharges of
events occurring in April 2021 relative to the latest gauging station discharge rating. However, the application of the learning
CVSG discharge rating after 12 months in hindsight demonstrated a stronger agreement to the gauging station rating in this

series of events. While the original learning estimate may have been correct at the time of the measurement, with non-stationary
site conditions resulting in a shift in the true discharge rating after a further 10 months of site evolution, it is considered more
likely that the lower agreement between the real-time CVSG learning discharge estimations and the latest gauge discharge
rating was due to discontinuities from the sparser CVSG device sampling during this period beyond the range of previously
recorded measurements.



**Figure 6: Correlation plots for the latest gauging station rating discharge timeseries against the CVSG estimated discharge timeseries at Tyenna River, Tasmania, Australia using (a) stereophotogrammetry estimated water levels, and (b) and gauge water levels, as well as at Paterson River, NSW, Australia using (c) stereophotogrammetry estimated water levels, and (d) gauge water levels.**

The Paterson River site demonstrated more challenging conditions for continuous optical measurements of surface velocities due to surface texture combined with CVSG device distance to the water surface at low flows, and the wider channel cross-section approaching the eye-level of the camera at the higher observed flow events. The resulting CVSG discharge estimations at Paterson River (Figure 6c/Figure S7a and Figure 6d/Figure S7b) demonstrate the utility of the learning model fit and discharge rating for improving the operational gauging capability using optically measured surface velocities.

Similarly to the findings at Tyenna River, the difference between the CVSG learning results relative to the latest gauging station discharge rating at Paterson River implied the majority of discharge gauging noise was the result of noise in the



stereophotogrammetry estimated water level (Figure S8a). This noise, as evidenced by the distribution of CVSG water level error (Figure 5c), was significantly worse at Paterson River during low flows approaching the upper range of the point cloud distance to reach the near bank intersection with the water surface. Despite this significant noise, the CVSG learning discharge estimations demonstrated a capability to reduce this influence of this error both in real-time learning, and improving further

with more measurements. Interestingly, the magnitude of raw CVSG discharge estimation errors was remarkably similar between the remotely sensed and gauge water level cases, but the reduced water level estimation noise when using the gauge water level (Figure S8b) displayed significantly reduced error in the CVSG learning discharge estimations converging much faster between the real-time and 4-month hindsight rating estimates.

## 3.4 Adaptive learning discharge rating

The latest CVSG learning discharge ratings after 12 months of deployment at Tyenna River using the stereophotogrammetry estimated water levels and gauge water levels were found to be within the range of historical manual gaugings (Figure 7). With the manual gaugings classified by decade, it is important to understand the foundation of the measurements behind the latest gauging station discharge ratings, as well as how degree of fit to the smooth gauging station rating curve. Whilst the majority of the CVSG learning rating using stereophotogrammetry estimated water levels were found to be within the range of the 344

manual gaugings undertaken at the Tyenna River site since the 1960s, the discharge ratings for stereophotogrammetry estimated water levels below 0.4 m (where water clarity impacted measurements) demonstrated a bias to higher discharge. There was also a transition point from high discharge estimates to low discharge estimates using the stereophotogrammetry estimated water levels in the vicinity of 0.6 m. Furthermore, the CVSG learning discharge rating using gauge water levels demonstrated a tighter convergence to the most recent manual gaugings and the latest gauging station rating.



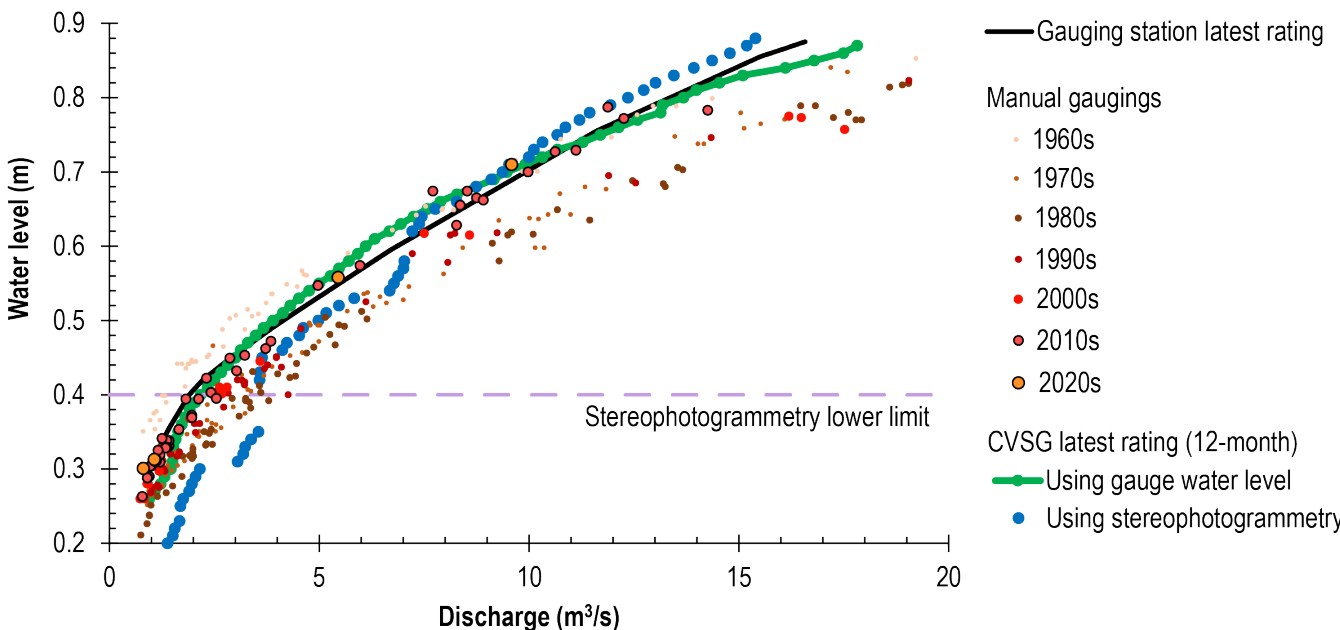

**Figure 7: Discharge gaugings at Tyenna River, Tasmania, Australia with CVSG learning ratings after 12 months of deployment using stereophotogrammetry estimated water levels and gauge water levels.**

It is important to note the upper range of discharge measurements above 0.79 m gauged by the CVSG system to 0.87 m has not been manually gauged at Tyenna River for more than 10 years where the RMSE to the latest gauging station discharge rating is above 2 m³/s with mean differences greater than 20% (Table 6). All real-time CVSG discharge estimations over the timeseries demonstrated less RMSE than the most recent decade containing an increased manual gauging range that includes the range measured by the CVSG system. For the stereophotogrammetry estimated water level-based CVSG discharge ratings, the RMSE showed an increasing trend with the progression of the learning process primarily due to an increasing gauged water level range containing sparse observations with more absolute discharge error. However, the NSE of these learning CVSG discharge ratings showed increasing skill relative to the latest gauging station discharge rating. The CVSG discharge estimations using the gauge water level demonstrated a relatively stable fit to the reference discharge rating, while the learning method was able to provide a significant reduction in the mean difference to the reference discharge rating.





**Table 6: Summary of site gauging results for Tyenna River, Tasmania, Australia relative to the latest gauging station discharge rating.**

| Measurement | Type | N | Range (m) | RMSE (m³/s) | NSE |
|---|---|---|---|---|---|
| Manual gaugings | 1960s | 57 | 0.35-1.21 | 1.11 | 0.964 |
| | 1970s | 82 | 0.25-1.33 | 2.79 | 0.916 |
| | 1980s | 90 | 0.23-1.51 | 3.23 | 0.884 |
| | 1990s | 43 | 0.27-1.54 | 2.58 | 0.936 |
| | 2000s | 25 | 0.26-1.40 | 2.18 | 0.823 |
| | 2010s | 42 | 0.26-0.79 | 0.38 | 0.989 |
| | 2020s | 5 | 0.30-0.71 | 0.32 | 0.990 |
| CVSG (stereophotogrammetry estimated water level) | Raw **(timeseries)** | 2133 | -0.33-1.05 | 1.05 | 0.904 |
| | Learning model fit **(timeseries)** | 2232 | -0.33-1.05 | 1.54 | 0.793 |
| | Learning rating **(timeseries)** | 2232 | -0.33-1.05 | 1.50 | 0.805 |
| | Learning rating 1-month | 1795 | -0.33-0.79 | 1.28 | 0.891 |
| | Learning rating 2-month | 2133 | -0.33-1.05 | 2.02 | 0.922 |
| | Learning rating 12-month | 17178 | -0.33-1.07 | 2.54 | 0.956 |
| CVSG (gauge water level) | Raw **(timeseries)** | 2141 | 0.31-0.87 | 0.63 | 0.961 |
| | Learning model fit **(timeseries)** | 9049 | 0.31-0.87 | 0.56 | 0.976 |
| | Learning rating **(timeseries)** | 9293 | 0.31-0.87 | 0.53 | 0.979 |
| | Learning rating 1-month | 3276 | 0.31-0.73 | 0.26 | 0.993 |
| | Learning rating 2-month | 3905 | 0.31-0.87 | 0.90 | 0.955 |
| | Learning rating 12-month | 19005 | 0.26-0.87 | 0.60 | 0.983 |

After 4 months of CVSG operation at Paterson River, the system had gauged the maximum water level range that had been manually gauged since as far back as the 1980s, with the CVSG discharge estimations using the gauge water level producing an estimated discharge at the top of this range within 0.22% of the discharge measured in the 2000s at this water level. Whilst
the manually gauged discharges at the upper recorded water levels appear to be sparsely measured, the CVSG discharge learnings using either gauge water levels or stereophotogrammetry estimated water levels were in agreement with the manual gaugings across the measurement range (Figure 8). Further to this, the shape of the CVSG discharge ratings agrees more closely with the shape of the rating curve implied by the manual gaugings, rather than the smoother fit of the gauging station rating curve.





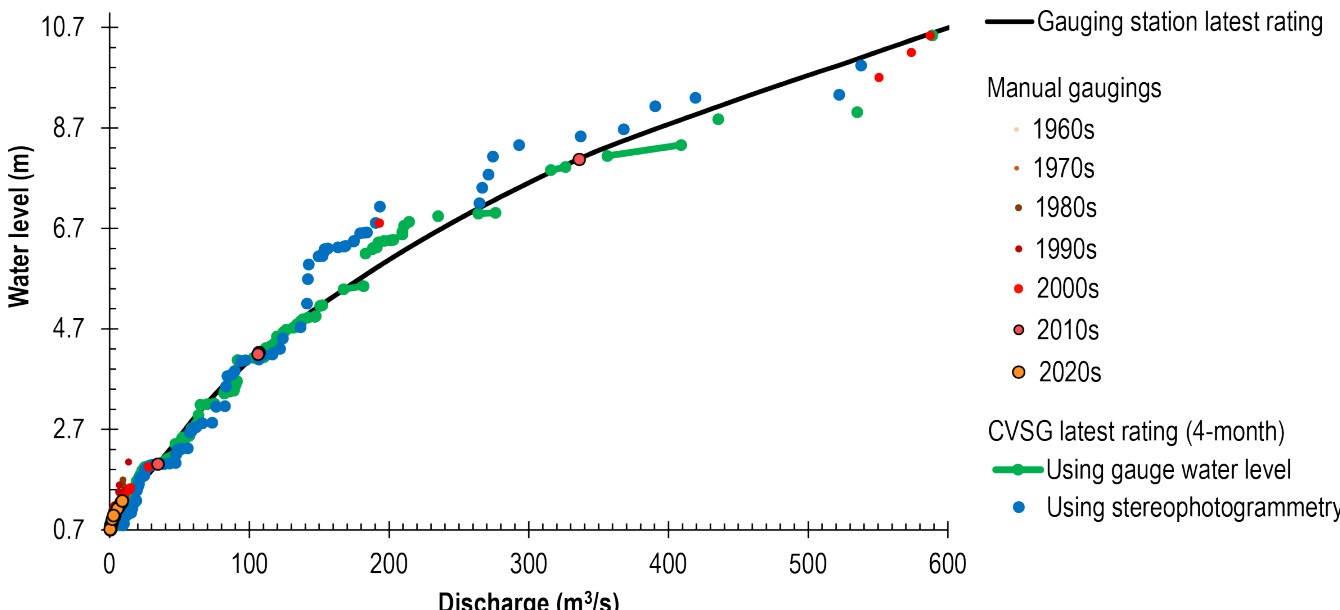

**Figure 8: Discharge gaugings at Paterson River, NSW, Australia with CVSG learning ratings after 4 months of deployment using stereophotogrammetry estimated water levels and gauge water levels.**

The summary statistics of the CVSG discharge estimations at Paterson River (Table 7) showed that the learning CVSG analysis using stereophotogrammetry estimated water levels were significantly impacted by the error in the water level estimation resulting in mean differences in excess of 100% relative to the reference latest gauging station discharge rating. However, the NSEs of the learning discharge estimations were significantly higher than the raw CVSG discharge estimations.

  





**Table 7: Summary of site gauging results for Paterson River, NSW, Australia relative to the latest gauging station discharge rating.**

| Measurement | Type | N | Range (m) | RMSE (m³/s) | NSE |
|---|---|---|---|---|---|
| Manual gaugings | 1980s | 17 | 0.81-1.69 | 4.45 | 0.455 |
| | 1990s | 45 | 0.66-2.04 | 4.51 | 0.540 |
| | 2000s | 50 | 0.56-10.54 | 29.42 | 0.917 |
| | 2010s | 38 | 0.61-8.07 | 6.20 | 0.908 |
| | 2020s | 7 | 0.71-1.28 | 0.24 | 0.993 |
| CVSG (stereophotogrammetry estimated water level) | Raw **(timeseries)** | 6228 | -0.21-9.94 | 21.92 | 0.680 |
| | Learning model fit **(timeseries)** | 6246 | -0.21-9.94 | 16.13 | 0.827 |
| | Learning rating **(timeseries)** | 6246 | -0.21-9.94 | 15.86 | 0.832 |
| | Learning rating 1-month | 1940 | -0.14-8.13 | 17.74 | 0.953 |
| | Learning rating 2-month | 3534 | -0.19-8.13 | 20.83 | 0.939 |
| | Learning rating 4-month | 6265 | -0.21-9.94 | 25.18 | 0.950 |
| CVSG (gauge water level) | Raw **(timeseries)** | 6592 | 0.78-10.54 | 19.04 | 0.973 |
| | Learning model fit **(timeseries)** | 18352 | 0.78-10.54 | 5.47 | 0.951 |
| | Learning rating **(timeseries)** | 18624 | 0.78-10.54 | 4.89 | 0.961 |
| | Learning rating 1-month | 1978 | 0.78-7.86 | 9.60 | 0.981 |
| | Learning rating 2-month | 3627 | 0.78-7.86 | 8.95 | 0.982 |
| | Learning rating 4-month | 6627 | 0.78-10.54 | 20.78 | 0.942 |

## 4 Discussion

The study has outlined a new non-contact optical stream gauging approach (CVSG) and provided a detailed comparative analysis as against existing optical approaches (DischargeLab and Hydro-STIV) and the current standard technologies and historical measurements informing gauging stations. Results highlight the advancements in stereophotogrammetry and
620 machine learning approaches can overcome some of the challenges of non-contact optical stream gauging and provide insights into the added value of these emerging operational technologies. Whilst a significant uncertainty on the order of 6% has been found to be possible from user variations in ADCP discharge estimations (Despax et al., 2019) as well as comparable deviations evident in a study of ADCP measurement validation (Oberg and Mueller, 2007), a similar order of magnitude in difference between direct technology comparisons was found in all-but-one exceptional case in an irrigation channel, NSW, Australia. A
625 study in an irrigation canal using an experimental video camera system with LSPIV and optical water level detection on a staff gauge demonstrated results within 5% of the reference estimates, but highlighted that reflections and shadows produced



negative effects on the detection of motion, requiring further consideration of spatial filters and light distribution (Lee et al., 2010).

There has been some LSPIV discharge uncertainty estimations undertaken from recordings of a stream in the French Alps (Dramais et al., 2011) showing less than 2% discharge deviation for recording durations more than 4 seconds, consistent with the CVSG results tested using different durations. With in situ profile measurements, this study in the French Alps found the uncertainty in the mean velocity coefficient to be close to 7%, with the largest source of uncertainty up to 15% possible without velocity depth profile measurements within the flow range of interest. Further to this, the sensitivity analysis of this French Alps study examined the effect of waves (negligible), cross-section transects (±4%), and water level errors at 4% from a 10 cm error (noting CVSG stereophotogrammetry water level error was within this range only 27% and 14% of the time at the Tyenna River and Paterson River sites respectively).

By using UAV footage of artificially seeded low flow conditions (average surface velocities between 0.12 to 0.14 m/s) on a river in Serbia, a significant sensitivity to algorithm parameters was apparent for LSPIV, large-scale particle tracking velocimetry (LSPTV) and optical tracking velocimetry (OTV) relative to the less sensitive SSIV (an enhancement of LSPIV) and Kanade-Lucas Tomasi image velocimetry (KLT-IV) (using an optical flow algorithm, similarly to CVSG) (Pearce et al., 2020). Whilst there has been some evidence that KLT optical flow performance degrades in low lighting (Wang and Miao, 2010), the Farneback optical flow approach used by CVSG has been found to provide more robust results in comparison studies (Nemade and Gohokar, 2019; Shi et al., 2020). This identified sensitivity of algorithm parameters has been reinforced by simulations using LSPIV showing substantial care must be taken for ensuring reliable results with regards to seeding shapes/sizes/densities, frame rates, recording durations, and camera angles (Hauet et al., 2008; Pumo et al., 2021). The general vulnerability of optical surface velocimetry methods to measurement setup (Detert, 2021) and environmental conditions, such as evidenced by a study of varying rainfall intensity particularly finding LSPIV-based autocorrelation methods to be susceptible to biased results under higher rainfall intensities (whilst results were demonstrably improved by sufficiently low rainfall intensities) (Naves et al., 2021), reinforces the need and utility of methodologies, such as those presented in this work, for increasing robustness to visual environmental noise pertaining to individual measurements.

The CVSG learning surface velocity distribution and discharge rating results evaluated in this study provide evidence to the benefits of the presented methods for improving measurement accuracy and reliability over time. Given increasing frequency of extreme rainfall are yielding flow events that exceed manual gauging records (Steinbakk et al., 2016), the ability of the CVSG approach to learn and adapt over time is particularly valuable. Whilst the entire range of the discharge ratings developed and evaluated did not necessarily yet contain a sufficient combination of quantity and quality of observations, this is similarly evident in the best available estimates provided from the reference gauging station discharge rating fits (which may be overly smoothed due to limited temporal and vertical-spatial data density). With the persistent measurement of velocities, sudden or gradual changes in velocity distributions over time can be detected in order to identify when the resurveying of a site's bathymetry is necessary (Peña-Haro et al., 2021).



The results of this study between CVSG estimated timeseries discharges and discharge ratings using stereophotogrammetry estimated water levels and provided gauge water levels, showcased the improvement in non-contact measurements that could be possible with improvements to the remotely estimated water levels alone. A combination of optical approaches to water level estimation using water line detection spatio-temporal texture histograms (Eltner et al., 2018) or deep learning (Eltner et al., 2021), as well as grayscale brightness, or motion segmentation (Peña-Haro et al., 2021) combined with a stereo camera

approach has potential for reducing remotely estimated water level noise and improve the range of suitable operational environments. This concept could simultaneously reduce the uncertainties and site deployment barriers associated with ground control reference points (Le Coz et al., 2021). Stereo cameras have demonstrated potential added value to stream gauging applications (Ran et al., 2016; Li et al., 2019), with the CVSG system providing a reduction in the barrier to the deployment of a stereo camera-based optical stream gauging site. With continuous automated site resurveying with every video recording,

the quantification of cross-section changes and vegetation growth offers significant advantages to monitoring streams without stationarity assumptions (Westerberg et al., 2011).

Whilst optical-based non-contact stream gauging has well-documented advantages for gauging high flow events compared to alternative methods (Le Coz et al., 2010), well-understood and relatively stable low-end discharge ratings could be provided to the CVSG system as a manually provided and updated basis for estimating low flows where low flow site conditions may

be unsuitable. However, the depictions of discharge ratings evaluated in this study are simplified one-dimensional water level dependent discharge estimates (as is commonly developed and applied for gauging station discharge ratings), but this is a simplification of the variation in discharge resulting from the changing hydraulic gradients across the rising and falling stages of hydrographs (Fenton, 2018). This nuance could be developed automatically through the significantly improved gauging data density offered by optical surface velocimetry approaches such as CVSG, providing an additional dimension for the

generation of discharge ratings dependent on both water level and the hydraulic gradient.

While the CVSG system evaluated utilised Savitzky-Golay filtering, it is noted that recent improvements on this method have been developed addressing the known pitfalls of the technique (Schmid et al., 2022). Beyond the approaches in this study for optically discerning water surface motion, further enhancement of optical measurement capabilities could be achieved through novel techniques using well-studied and bounded principles of fluid dynamics (Khalid et al., 2019). An ensemble of surface

velocimetry techniques could be applied, given sufficient computational power, to provide an additional quantification of algorithm-based uncertainty similar to ensemble approaches employed in other fields for quantifying model structural sensitivity (Nearing and Gupta, 2018), and facilitate the identification of disagreements between methods under particular sites and conditions over time while expanding the broader applicability of the technology through the advantages offered by each technique. Furthermore, the optical nature of the methods developed supposes the possibility for the incorporation of additional

computer vision analysis through rainfall (Jiang et al., 2019; Chen et al., 2019; Wang et al., 2022), wind (Cardona, 2021), and water turbidity (Leeuw and Boss, 2018) monitoring.




## 5 Conclusions

This study has demonstrated the successful development of an automated operational optical stream gauging system employing methods providing improved reliability for remotely gauging streams using state-of-the-art surface velocimetry technologies

across varying flow and lighting conditions. Evaluation of the existing best practice in available stream measurement technologies and published discharge ratings across the array of site conditions evident in this work demonstrated that the methods in this study were similarly effective for gauging stream discharge to existing accuracy benchmarks. Additionally, whilst the technology demonstrated in this study is well-developed for operational use providing added value and capabilities over other measurement technologies that have been established for some time, there is potential for significant future

developments which have been identified for further improving the utility of the approaches presented. The non-contact and automated solution offers a significantly greater density of velocity-stage observations resulting in up-to-date adaptively learning discharge ratings through time. As climate-driven extreme weather events increase in frequency, it is increasingly important to develop and apply flexible monitoring tools, such as CVSG, that can reduce the human and environmental risks associated with traditional approaches and deliver real-time data to water resource managers.

**Code availability**

Code not available at the time of publication

**Data availability**

All raw data can be provided by the corresponding authors upon request.

**Author contribution**

NH, RB, DW, JS, ND, AG, and SA conceptualised the work; NH, RB, and DW curated and analysed the data; DW, JS, AG, and SA contributed to acquisition of the funding supporting this work; NH, RB, DW, JS, ND, BE, AG, and SA contributed to the investigations of this work; NH, RB, ND, BE, AG, and SA contributed to the methodology development; NH, RB, DW, JS, BE, AG, and SA contributed to project administration; NH, RB, DW, JS, BE, AG, and SA contributed to the provision of the project resources; NH and BE contributed to the software development related to this work; JS, AG, and SA provided

oversight and leadership in the supervision of the work; NH, RB, and DW contributed to the validation of the research outputs; NH prepared the visualisations, and original draft of the published work; RB, DW, JS, ND, BE, AG, and SA contributed review and editing of the work.



**Competing interests**

The authors declare that they have no competing interests.

**Acknowledgements**

The authors would like to thank James Mancey (Department of Natural Resources and Environment Tasmania, Australia) and Anthony Belcher (WaterNSW, Australia) for providing study sites and historical discharge gauging data. The authors would also like to thank Paul Blumer (Murrumbidgee Irrigation) for providing the example irrigation channel analysed in this study. This work was supported by Henry Valk's (IoTamy) and Nicolas Turner's (The University of Queensland, Australia) tireless
efforts in the early development of the CVSG device integrated hardware solution which collected a substantial portion of the data presented in this study. This research was supported by an Australian Government Research Training Program (RTP) Scholarship.

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
