# Peer review of "Adaptively monitoring streamflow using a stereo computer vision system"

_EGUsphere, 2022_

## Author Comment (AC1)

**Reviewer 1**

This is an interesting comparison of a variety of techniques for discharge estimation with a view to evaluating the CSVG stereophotogrammetry method for deriving discharges from surface velocity measurement, including the use of an (unspecified) adaptive learning algorithm. I do think, however, that the paper could be significantly improved, in part because the details of the CSVG method are kept almost deliberately vague as if to not give too much away (without actually saying so, though implied by the software code not being made available). However, this makes it really frustratingly difficult to understand what lies behind some of the results. It is suggested, for example, that the method can produce comparable discharge estimates to traditional rating curve and ADCP methods – but only really if a local water level measurement is available (not that this is really a problem these days when low cost methods are available). In the points below I have suggested many places where more detail is needed – if not in the paper directly then referenced to material in the supplementary file.

*Reply: We thank the reviewer for their time and detailed feedback, particularly with regards to improving the clarity of the methods. While we have not intended to provide deliberately vague details, we believe the size of the paper, including the weight of the methodology section and the scope of site data results, necessitated focusing less attention on the specific details of particular functions. We instead provided a balance of the detail required to describe the conceptual approach applied, and further to this detailing the most important components of the application of the approach. The software code is not available, as it is mostly comprising the integration of the hardware recording and analysis data structures with cloud databases and services embedded with the application of the methodology that is described for the analysis approach. We have answered the specific points below, including minor revisions based on valuable feedback from the reviewer for improving the clarity of the manuscript.*

**Some specific points are follows:**

I would suggest that the results are reordered somewhat so that each site is considered in turn as because the Paterson site is so different from the others – from the photo it would appear that this is only site where a downstream (rather than cross-channel) camera view is used with a flow that does not seem to have developed a uniform flow profile. The reasons for the failures here need more discussion (as shown in Fig S7).

*Reply: Thank you for your comment – we believe by your description that you are actually referring to the irrigation channel site with regard to the downstream facing camera view, whereas Fig S7 is from the Paterson site (as you say), which is shown in Figure 2b with a cross-channel camera view.* *We have now highlighted how different this site is from the others in the results section on L424.* *Furthermore, the description of the results in Figure S7 has been provided on L541.*

**Added to L424:** "It is important to note the irrigation channel site differs substantially from the other case study sites with a downstream field of view and highly turbulent flow conditions discharged through an engineered channel."

L83 para. Yes, but what technological advances do you mean? Those in the current study? Those to come (in which case more detail needed). Might be better moved to end.

*Reply: Thank you for the insight for this clarification and suggestion for improvement in the flow of ideas. This statement does refer to the technological advances as applied and tested in relation to the advances that facilitated this work. In light of your feedback, we agree and have moved the statement to the end of the introduction section.*

L118. Not clear how this 40m relates to the 10m on L156, and how the camera resolution and the 120 degree field of view create the 0.1m analysis resolution?

*Reply: Thank you for raising this question. The 40 m mentioned on L118 is relative to the position of the camera and referring to the system hardware limitation for estimating water levels using stereophotogrammetry requiring the water near edge to exist within the vision of the camera within a 40 m range from the camera's physical location, while the 10 m on L156 is describing the region of the water surface used for stereophotogrammetry water level estimation relative to the near bank interception with the water edge. The camera resolution and field of view has no relation to the 0.1 m analysis resolution, which is simply the fixed size of the grid that the estimations are projected on (while in practice this grid size is actually an adjustable parameter, all results presented and all deployments of the system have used this default analysis grid resolution of 0.1 m). Further explanation of the optical flow resolution calculation step has been added to L121 to describe how the fixed 0.1 m analysis grid resolution is not directly connected to the camera resolution (which has these camera hardware limitations accounted for prior to reaching the analysis grid).*

**Added to L121:** "The optical resolution of the flow in meters per pixel is calculated based on the water surface projection in order to filter any motions in the area of the field of view beyond the limits of acceptable optical flow resolution accuracy (normally limited to a maximum of 0.05 meters per pixel up to 0.2 meters per pixel)."

L124. What do you mean by adaptive learning (you also refer to machine learning later)? No details are given. And here you do not mention the issue of going from surface velocities to profile or mean velocities (see comment on L221)

*Reply: This section is intended as an overview of the system to describe all the aspects and how they relate to each other as well as how this relates to the practical use of the system. The adaptive learning refers to the process described in the later sections of the methodology in 2.3 and 2.4 which result in an adapting surface velocity distribution (adaptive to new observations if changes occur at a site) and learning from new observations to add to the database of velocity distributions which are then each calculated for contribution to the discharge rating. As for mentioning the issue of going from surface velocities to profile or mean velocities, this is a well-known and studied feature of all methods for estimating discharge based on observations of surface velocities, as described in the introduction and the procedure applied for this is detailed at the beginning of section 2.4. Machine learning is referred to generally as the method presented was built to leverage collected sample data to improve performance in gauging stream flows, hence demonstrating the potential for machine learning approaches to overcome challenges in optical stream gauging using cameras.*

L158. Why the first percentile (indeed what does the first percentile mean)?

*Reply: Using the first percentile in this context essentially allows you to quickly take the near-minimum without taking any sporadic outlying minimum value arising from*

*erroneous points in the generated point cloud. We have added this note by modifying the line at L158 to improve the clarity of the reasoning behind the choice of approach.*

**Modification of L158:** "The first percentile of the elevation points of the stereophotogrammetry cross-section profile within this domain is then estimated as the water level (effectively taking the near-minimum of the surface while reducing the impact of any sporadic point cloud artefacts)."

L168. What is this minimisation problem? Since it will affect the estimates it needs more explanation – at least in the supplementary file

*Reply: Thank you for your question and keen interest in the detail behind the algorithms used in this work. We do not think it is reasonable to reproduce this explanation in detail and would like to direct you to section 3.2 of the reference source material for the Farneback optical flow algorithm cited in this line. We have modified this section from L171 to add some detail of the Farneback algorithm for estimating optical flow and provide a reasonable summary of the background detail to the reader.*

**Modification from L171:** "Shi et al. (2020) compared three established and widely applied optical flow techniques to breaking surges, noting the advantages of the Farneback algorithm for its relatively high accuracy and dense flow fields, as well as a lower sensitivity to noise with the converging iterative solution for the displacement vector, $d$, between a pair of images using quadratic polynomials following Eq. (1):

$$d(X_{\text{im}}) = \left(\sum_{\Delta X_{im} \in I_{local}} wA^T A\right)^{-1} \sum_{\Delta X_{im} \in I_{local}} wA^T \Delta b_f , \tag{1}$$

where $I$ is the greyscale image with local neighbourhood regions denoted by $I_{local}$ using the image coordinates $x_{im}$ and $y_{im}$ to form $X_{im} = \begin{bmatrix} x_{im} \\ y_{im} \end{bmatrix}$, where the change in brightness between the corresponding pixels in the pairs of images are denoted $\Delta X_{im}$. Furthermore, $w$ is a weighting function over the local neighbourhood regions, while the polynomials are defined by $f(x_{im}, y_{im}) \cong a_1 + a_2 x_{im} + a_3 y_{im} + a_4 x_{im}^2 + a_5 y_{im}^2 + a_6 x_{im} y_{im}$ with $A = \begin{bmatrix} a_4 & \frac{a^6}{2} \\ \frac{a^6}{2} & a_5 \end{bmatrix}$, $b_f = \begin{bmatrix} a_2 \\ a_3 \end{bmatrix}$, and $c = a_1$.

The approach is a variational method combining the assumptions of local neighbourhood brightness intensity variation between frames with the minimisation of an energy function assuming a slowly varying displacement field for locally smooth velocity gradients (Shah and Xuezhi, 2021)."

L187. Motions out of the water surface? Some hint here of a limitation but these are on a surface, needs more explanation. And filtered how? As NANs, or with some replacement strategy?

*Reply: While we very much appreciate your excellent reviewing mindset towards finding potential limitations, and we appreciate that you would be aware of the many limitations present in the available/established methods for measuring natural open channel stream flows, the key motivation for this step of the procedure is the removal of motions that are optically visible to the camera, but are not part of the measurement of the planar surface velocity contributing to the measurement of the nett discharge of water through the stream section. As such, the concept of filtering is used in the regular sense of the word where unwanted material (motions in the vector field which are out of the plane of the assumed water surface) is removed without any other replacement strategy that isn't already described in the methods. We have added the word 'assumed' for clarity to L187.*

**Modification of L187:** "From this point, the motions out of the assumed plane of the water surface are filtered out of the analysis to further remove false motions unrelated to the waterway surface velocities (such as animals and swinging ropes which are not moving in the assumed plane of the water surface)."

L190 Continuity of streamlines imposed how?   What assumptions about the nature of the streamlines?

*Reply: Thank you for this comment, we agree that statement is not entirely clear. We have modified this section from L189 to be more descriptive of the assumption of the nature of the flow over the analysed section.*

**Modification from L189:** "Assuming the remaining velocities over the length of the analysis section are velocities related to the motion of the water surface, and assuming a continuity in the uniformity of the analysis section length without transitional flows, the strongest detected velocities are collapsed into a single-dimensional raw cross-section surface velocity profile. The assumed continuity over the analysis section length facilitates the measurement of velocities across spatially inconsistent optical flow measurement/lighting conditions along the length of the analysed section."

L198 "multiple measurements of the same water level over time in different conditions to.combine these measurements into a complete velocity profile" – totally obscure.   Different measurements at the same water level should give you an estimate of variability of estimates at that water level, but why does it tell you anything about the profile.   In fact you do not seem to consider the profile at all – only using data from elsewhere to estimate a coefficient to convert to mean velocity.

*Reply: We agree that this is confusing, and have attempted to take great care to use the words profile and distribution as clearly as possible when referring to velocities over the stream section. The intent of this statement was not to say anything about the profile of velocities beneath the water surface (which, as you know, are not directly measured by this approach), but to instead refer to the profile of surface velocities across the cross-sectional profile from one side of the stream to the other. We have added the descriptor modification in L197 to make clear that the velocity profile being referred to is the surface velocity profile.*

**Modification of L197:** "This process of developing an adaptive database of surface velocity measurements across the stream at different water levels (adaptive learning surface velocity distributions), allows the system to use multiple measurements of the same water level over time in different conditions to combine these measurements into a complete surface velocity profile, while simultaneously being adaptive to observed changes in surface velocity profiles in non-stationary environments."

L204. Why exponential?   Are there not theoretical 2D cross-sectional distributions that you could have tried (though presumably would not be valid for the Paterson site).   And in fitting the distribution, what if it is the highest values that are not available?

*Reply: The exponential relationship from the boundary distance factor to the surface velocity profile is just a simple logarithmic relationship of the rearranged form -bx = ln(1-V_s/V_infty). Many alternative relationships and boundary distance factor transformations were tested against optically estimated surface velocity distribution observations at different sites, but the form presented here was the best generically fitting to the data. Thankfully due to the nature of the optical approach, the highest surface velocity values in the distribution are ordinarily the most available owing to a high signal to noise ratio, but cases can arise where these highest values are out of view or obscured by vegetation or other visibility challenges. If the rest of the distribution is intact, then V_infty will be fitted based on the trending asymptote of the observed surface velocities in the transformed boundary distance factor domain. However, if the distribution of the surface velocities is not well-enough observed, then in any case you cannot very well predict the discharge using surface velocimetry unless*

*you have learned these through previous observations or surrounding observations at different water levels (as are both included aspects of the approach presented).*

L221 – should not values of a be considered uncertain (and should this uncertainty not be propagated into the discharge estimate (see the cross-section you show in Figure S3)

*Reply: We agree that this is uncertain, and is why we keep track of an envelope which is mentioned in your next point about L230. In fact, we would generally agree that all discharge estimates should be reported as and thought about in terms of estimated ranges (as an indirectly quantified measure). While standard parameters for the calculation of 'a' are configured as part of the analysis configuration, a standard +- 15% 'a' range is applied to the independently learning minimum and maximum surface velocity (and hence discharge) envelope boundaries provided with the data reporting. This discussion is then continued in the next point.*

L230.  You do not say where these adapted learning distributions come from (and should that not also be associated with an uncertainty estimate using e.g. Bayes updating).  You mention an "envelope" but that never appears later in the results.

*Reply: The authors appreciate that the origin of the usage of the term adapted learning (surface velocity) distributions may be unclear, and confirm that the adapted learning distributions indeed come directly from observations made by the system aggregated from different points in time where the same water level has been measured as explained in the previous section from L196 onwards and exampled in the supplementary Figure S3. We have added the identifying term in L198 to make this connection clear and improve the clarity of the manuscript thanks to the reviewer's feedback. The authors have decided to leave out the "envelope" results (which are generally enveloping of all available estimates) to avoid cluttering the results, and focus on the differences between the best estimate provided (particularly given that a single number is ordinarily taken as the best estimate from the gauging stations compared).*

**Modification of L197:** "This process of developing an adaptive database of surface velocity measurements across the stream at different water levels (adaptive learning surface velocity distributions), allows the system to use multiple measurements of the same water level over time in different conditions to combine these measurements into a complete surface velocity profile, while simultaneously being adaptive to observed changes in surface velocity profiles in non-stationary environments."

L237.  We do not need quality codes – we need proper uncertainty estimates.  You surely have the information to be able to do so.

*Reply: The authors agree with the reviewer around the need for proper uncertainty estimates, however we consider what constitutes a 'proper' uncertainty estimate to be reasonably debatable, and this paper does not yet seek to present techniques or support of any particular set of uncertainty estimation approach in connection with this work beyond the methodology detailed. The authors do not agree with the comment about quality codes, as this is a data documentation approach required and applied in practice by water agencies across Australia and internationally. We have added to the discussion at L636 on uncertainty estimation methods with reference to a recent comparison study (Kiang et al., 2018).*

**Added to L636:** "Alternative methods for estimating the uncertainty of stream discharge rating curves have been compared in Kiang et al. (2018), finding a wide variation in uncertainty estimates resulting from different methods which demonstrated the necessary careful selection and communication of the assumptions of the uncertainty estimates provided."

L261. Why NSE? That seems inappropriate for a rating curve since NSE scales by the observed variance which is here over the depth values). That is more like a regression so is not a correlation coefficient more appropriate?

*Reply: We appreciate the reviewer's feedback and point of view, but we believe that a correlation coefficient is not more appropriate than the NSE as a statistical metric in this circumstance. Whilst NSE is most notably applied as a skill metric for the fit of hydrological flow timeseries data due to the way it is less skewed by the more frequently observed and perhaps (depending on study objective) less important (and ordinarily easier to predict somewhat closely) low flow data relative to the less frequently occurring flow events which have more significant error margins. In this regard the NSE is more sensitive to extreme values, and the authors consider this to be important for appropriately assessing the rating curves as they have been likewise constructed through time by many data points towards the lower end of the discharge scale and fewer observations towards the higher end (which is also similarly the case for the manually gauged observation distributions). We have added this reasoning in brief to the end of this line at L258.*

**Modification of L258:** "At two existing government maintained gauging stations, historical manual gaugings have been compared along with CVSG, DischargeLab, and Hydro-STIV measurements relative to the latest published discharge rating using root-mean-square error (RMSE), the mean percentage difference, and the Nash-Sutcliffe Efficiency (NSE) (Jackson et al., 2019) commonly applied for assessing predictive skill for discharges in hydrological settings due to its sensitivity to extreme values."

L280 Table 1 – there seem to be some inconsistencies in presentation here (e.g. water levels of 135m and 0.31m are clearly not both relative to local datum?)

*Reply: Thank you for this comment, the authors recognise that the relevancy of the water levels presented is not in the absolute value, but rather the range of water levels over which observations occurred. In light of this, we have added the note that these water level ranges have been presented relative to local datums in the Table 1 caption while pluralising the relevant Table 1 heading.*

**Modification from L280:**

**Table 1: Field case study sites summary (water level ranges presented relative to local datums).**

| Site | Period | Distance to stream (m) | Water levels (m) | Reference gaugings | Ground control reference points |
|---|---|---|---|---|---|
| **Castor River, Ontario, Canada** | 30 s | - | 3.77 | 1 concurrent (2019) | 12 |
| **Irrigation channel, NSW, Australia** | 30 s | - | 135.80 | 1 concurrent (2020) | 10 |
| **Tyenna River, Tasmania, Australia** | 56 d | 5.9–7.3 | 0.31–0.87 | 344 historical ('64 – '22) | 9 |
| **Paterson River, NSW, Australia** | 122 d | 0-22.5 | 0.78–10.54 | 157 historical ('87 – '21) | 0 |

L311. More detail needed on the ADCP for clarity– was averaging over multiple transects or other filtering of anomalies down

*Reply: Thank you for this feedback towards improving the clarity of the manuscript. This detail has been added from L305.*

**Modification of L305:** "A 30 Hz 30-second video recording (3840x2160 pixel resolution) formed the basis for the surface velocimetry estimations, with a reference measurement provided by a series of four SonTek RS5 moving boat ADCP (San Diego, CA, USA) transects taken between 15 to 20 m downstream of the hydraulic control structure within a timespan of eight minutes and a maximum discharge estimation difference of 8.5% to the most outlying transect measurement."

L373. But Figure 3 does not really support this – either there appears to be little difference or for Paterson it seems disadvantageous.

*Reply: This line forms part of the overall results introduction which summarises the results section in its entirety before detailing the results of each section. The previous line of this section is applicable to Figure 3 with the caveat of 'under suitable conditions' noted.* *The authors have expanded this line further to clarify the subject of this statement.*

**Added to L370:** "The results of this work found broadly comparable gauging results using the raw data of the different measurement technology approaches employed, predominantly falling within a relative error of 15% under suitable conditions when comparing between the results of both the detailed surface velocity distribution case studies and longer deployment timescales evaluated."

L373. But is it not the large percentage that is greater than 0.5 m that is more significant (as clear in Figure 5)? It is unclear why a stereophotogrammetry method can be >0.5m in error for so much of the time. Is this a result of the particular camera system used? It is off the shelf but has only 120mm separation between the lenses.

*Reply: Figure 5 presents the percentage of water level error <0.5 m, whilst the >0.5 m error percentages are the remaining 2% and 38% for the Tyenna River and Paterson River sites respectively. We are not sure if the reviewer has personal experience with measuring water levels from a distance in natural riverine environments, or if the reviewer is aware of any previously published and evaluated datasets with comparable distances and timescales, but these are the results of the particular camera system we used while applying the method described to estimate water levels through stereophotogrammetry. Since this data was analysed, further development has improved this accuracy somewhat using a calculated ambient environment correction factor, but the results presented in this manuscript were produced and analysed prior to this additional development.*

L425. Well yes (look at the photo)! So should you not present this as a "test to failure" type of site? You would not actually have had to go much further downstream to have been more successful.

*Reply: We are pleased that the reviewer agrees with the statement made by the authors. However, moving the site downstream would represent a different site, which is not the site of the case study.* *Figure S1 in the supplementary materials has been revised demonstrating the relative locations and discharge estimations resulting between each of the methods along the section reach length.* *The authors previously described the optimisation of the analysis region for each measurement technology's*

*most suitable region of interest reaching the best available result for the irrigation channel site in the methodology section on L314.*

**Modification of Figure S1:**

[Figure]

**Figure S1: Raw discharge measurements using different technologies along the length of the irrigation channel in NSW, Australia.**

L470.  This appears to be a combination of trend in cross-section/rating as well as statistical observational variability for that depth.   So when you refer to the "latest rating curve" – what period of observations is used to define that curve?   (Also Figs 7,8, Table 6, etc later)

*Reply: The authors appreciate the reviewer's observation of this fact. We would like to clarify that the latest gauging station rating fit refers to the best estimate published by the government agency using a best fit of past manual gaugings by a professional hydrographer (representing the best available estimate using the currently applied technology and data for each site). We have taken care to add the necessary*

*clarification detail to what this 'latest rating curve' means in the methodology from L255.*

**Modification from L253:** "Additionally, historical ADCP derived estimates of discharge used to develop discharge ratings were utilised as a reference. Whilst the most up to date discharge rating fits published by government agencies based on the professional judgement of hydrographers using the applied technology and data available prior to the deployment of optical methods at each site were used to represent the best available estimates."

L502 Why do you refer to correlation plots without giving correlation coefficients?

*Reply: We do not feel that correlation plots necessitate the presentation of correlation coefficients unless this provides a relevant insight. The error has been broken down visually and quantified into different classification groups, presented in a way that the authors believe is more insightful and relevant to the context of the problem and data being evaluated. If there is a relevant reason for providing these coefficients, then the authors can add these to the figure or figure caption.*

L521What do you actually mean by learned discharge rating curve? Is it purely a filtered estimate over time that will average out error, or is other data input to the process (you have not said how it works earlier). Clearly if you input the actual levels (or weight by error relative the the measured level) you are going to get much closer to the "latest rating curve" as shown in the other plots (and Figs S5, or even S7).

*Reply: The learned discharge rating curve is described earlier in the methodology section 2.4 (titled 'Adaptive learning discharge rating'). It has been applied precisely in the way it is explained by leveraging a fit across all of the adaptive learning surface velocity distributions that is described in the methodology section 2.3 for each of the observed water level increments. There is no additional input of actual (gauging station measured) water levels to assist in improving the discharge ratings derived from the stereophotogrammetry estimated water levels. The use of gauging station measured water levels is only applied completely independently in-place of stereophotogrammetry estimated water levels where the discharge from each are compared for evaluation purposes.*

---

## Author Comment (AC2)

**Reviewer 2**

The authors present an interesting and novel hardware/software package for computing river streamflow using camera imagery. In their package called the Computer Vision Stream Gauging (CVSG) system, they deploy a stereoscope camera paired with an edge computing device to capture stereo video in order to process into velocities, cross sectional geometry, and water surface plane determination. The CSVG is unique in that it does not require ground control points to perform image velocimetry analysis. Also, the CVSG uses a machine learning technique to improve results over time at a site as the equipment is allowed to collect more measurements. The CVSG seems to perform at least as well as other commercially available image-based velocity software. In one particular case, the CVSG completely outperforms other techniques, in part because it can interpret complex water surface profiles, such as cases with standing waves or other extreme conditions which normally highly degrade other common image velocimetry approaches. The authors do a good job evaluating the performance of the CVSG velocity and dishcarge results against standard/conventional methods and show excellent agreement. Where the system performs worse than conventional methods, the authors explain the limitations of the system. The authors should consider explaining a bit more about the approach for computing learned water surface plane and cross-sectional shape with the stereoscopic imagery. As with other non-contact streamflow methods, the detection of error in cross-sectional error is difficult. Since the CVSG system can presumably develop information about the cross-sectional geometery owing to the steroescopic camera approach, the authors may consider expanding the work a bit to describe, if applicable, the performance of the CVSG to determine or evaluate changes in cross-sectional area compared to a priori knowledge (e.g., input surveys of the cross-section). Otherwise, I believe this manuscript describes a useful advance in non-contact streamflow techniques.

*Reply: We thank the reviewer for their time and feedback about the manuscript. The authors agree that there is significant untapped potential in the rich amount of spatial data that is generated from the stereoscopic imagery that is computed where the current implementation (as described) only begins to scratch the surface of what is possible. Detail has been added about the simple approach for the site learning cross-section in section 2.2 from L150, however none of the sites or data presented made use of this aspect and this fact has been similarly clarified (all were provided with fixed manually surveyed cross-section profiles that were not free to be adapted to the ongoing live surveying of the terrestrial profile provided by the stereophotogrammetry). We believe this remains an important area for future research and evaluation requiring studies over longer timespans with ongoing manual surveying comparisons. We have added this important note to the discussion section at L681.*

**Added from L150:** "While the CVSG system maintains an adaptive cross-section database for each site which is compared and adapted with each measurement for visible terrain above the estimated water level (applying more weight to gradual changes in time and requiring many consistent measurements to gradually apply any observed dramatic changes in the cross-section profile), the results of this study applied fixed manual cross-section surveys from the time of deployment over the entirety of the time periods evaluated."

**Added from L671:** "A comprehensive study evaluating the use of stereo camera systems such as CVSG for quantifying adaptive cross-sections is an important area of future research to be determined over studies spanning longer timescales with significant erosion and/or accretion events at suitable study sites."

In addition to the comments above, I note several comments below, organized by line number. After consideration, I would happily recommend that the manuscript be accepted with some minor revisions.

Line 65: I agree. In part this is because there have been few available quality software products to aid in adoption by hydrometric agencies. This is starting to change, but it is a slow process.

*Reply: We thank the reviewer for sharing their view point on this matter and the authors agree with these comments as it is in agreement with our experience in discussion with various hydrometric agencies.*

Line 105: It would be very helpful to report the CVSG power consumption information. I would presume that the camera is a significant portion of the power budget.

*Reply: The authors thank the reviewer for this feedback.* *We have added details of the characteristic power budget of the deployed CVSG hardware at L107.*

**Added to L107:** "The total power consumption of the CVSG hardware collecting data in this study was on the order of 36 W hr per day, averaging 1.5 W with a peak power draw of 30 W."

Line 118: IMUs are notorious for drifting when position is fixed for a long period of time. Can you describe how the CVSG accounts for this? Do you observe IMU drift? Owning to the requirement of seeing a horizon line, are you correcting IMU drift from the horizon?

*Reply: We fortunately have not observed any notable IMU drift from any CVSG hardware installations after the initial factory calibration has been applied. We only use the linear acceleration in the x, y and z dimensions to determine the orientation of the camera for each measurement compared to the last known orientation. We have observed drift in the pose estimated by the same IMU under moving measurement applications, but the absolute measure of the orientation of the camera relative to the force of gravity has proven to be robust (even in moving applications). We do not require or use the horizon line for any corrections, as this would be quite restrictive for the selection of appropriate sites.* *We have clarified the wording from L139 to reduce any confusion about the horizon in the field of view. This modification is included in the modification to the same text in the next comment.*

Line 140: I am presuming that this requirement is to avoid glare? Can you expand on this statement?

*Reply: The authors thank the reviewer for this comment.* *The reviewer's presumption is correct, and we have added this expansion to L139 to improve the clarity of the reasoning for the guidance provided.*

**Modified from L139:** "When selecting a site, care should be taken to identify sites with suitable surface flow visibility and oriented south-facing (southern hemisphere) or north-facing (northern hemisphere) where possible to avoid sun glare, while keeping the horizon or sky outside of the camera field of view (maximising the water surface in the field of view and reducing automatic exposure determination from the sky)."

Line 141: So this would indicate a high-oblique view? As in the sky is not visible in the field of view? If so, does this eliminate the possibility of IMU calibration using the horizon line?

*Reply: This is correct that the sky is preferably not visible in the field of view from the perspective of maximising the field of view available for measurement of the stream surface while minimising any negative effects from autoexposure to the brightness of the sky.* *We have clarified the wording here*

*in L139 to reduce the possibility for this misunderstanding of the camera angle recommendation. This modification is included in the modification for the previous comment.*

Line 145: Can you provide details about how the stereoscopic camera determines the the land surface? What quality assurance methods are in use? What is the accuracy of the stereoscopic transformation...the parallax of the camera in use is fairly small, so I would expect that there is potentially significant errors in the transformation process. What about obstructed views owing to shadows, obstructions, etc. (e.g. boulders or even cobbles may present "shadows" unseen by the stereo camera ... what are the impacts to cross-section geometry accuracy from these sorts of artifacts?)

*Reply: The authors thank the reviewer for this suggestion to provide this detail. We have added this information to this section 2.2 at L151, significantly improving the description that was provided previously. For any obstructed views which are not unobstructed along any of the search lines within the cross-section analysis region, this would simply appear as missing data, and you would seek to avoid setting up the system at a site with obstructions like this without at least partially providing manual survey data covering the obstructed areas.*

**Added from L151:** "Stereophotogrammetry is applied to estimate the distance from the camera to features which are matched between the stereo pairs of rectilinear corrected images where a convolutional neural network model (provided by the camera manufacturer, Stereolabs), that has been trained on pairs of stereo images, is applied to improve both the accuracy and solution density particularly with reflective and featureless surfaces."

Line 161: Because you are using an IMU and stereo camera to determine the water plane, is there potential that the CVSG would be able to better handle high-slope systems, where the typical mono-lens camera approaches solve for a water surface plane that is oriented parallel to the Z coordinate (which respect to gravity)? This would be a useful differentiation between typical camera-matrix solution approaches for rectification vs epipolar geometric approaches using stereo cameras.

*Reply: The authors thank the reviewer for raising this point. We have added this note in a modification from L147.*

**Modification from L147:** "However, a stereo computer vision system also makes it possible to initially survey and then continuously monitor the terrain of the cross-section above the water level for changes due to erosion, deposition, or vegetation, and offers the potential advantage for measuring surface velocities on variable or steep hydraulic gradients."

Line 183: How is averaging of the flow field stack able to suppress motion artifacts? Wouldn't a median be better?

*Reply: We agree in principle that applying a median would be more idealistic for accurate surface velocity estimates across a wide range of conditions relating to optical flow visibility, and early tests trialled both approaches. However, in practise we have opted for the analysis of the presented results to continuously add each frame to an accumulating flow field and take the average by dividing the number of frames in order to reduce the computational hardware requirements for memory usage when scaling up to longer duration measurements consisting of many frames while maintaining compatibility with the existing edge computing hardware. This has now been noted as an addition from L183.*

**Added from L183:** "While taking the median of the flow fields would be reasonably more preferable in this context, the average accumulating flow field computation is applied to reduce the edge computing hardware

requirements of the method, particularly with memory usage as the duration of the measurement scales the number of instantaneous flow field frames stored in memory for a median calculation."

Line 187: Would would these erroneous motions indicate physically? For example, would camera motion (e.g., wind for example buffeting the instrument) be one of these extraneous motions?

*Reply: The authors have observed that the physical erroneous motions that are referred to in this statement relate to wind shaking vegetation or swinging ropes in the foreground of the frame, as well as bugs and animals that are not moving in the assumed plane of the water surface. Most oscillatory motions (such as the result of wind buffeting the instrument) are removed or reduced in the flow field accumulation step explained in a prior step of the methodology. We have added examples of these erroneous motions to L187.*

**Modification from L187:** "From this point, the motions out of the assumed plane of the water surface are filtered out of the analysis to further remove false motions unrelated to the waterway surface velocities (such as animals and swinging ropes which are not moving in the assumed plane of the water surface)."

Line 228: This is an improvement over the standard approach of one alpha value per section. I agree with this approach.

*Reply: We thank the reviewer for this comment, as we have found this to be a reliable approach to date.*

Line 245: Although I understand the scope of this paper is to compare the CVSG to other commercially available image velocimetry approaches, I think it would strengthen the work to also show results using some of the other well-adopted approaches in the literature, for example Patalano's RIVeR (https://doi.org/10.1016/j.cageo.2017.07.009) Perk's KLTIV (https://doi.org/10.5194/gmd-13-6111-2020)

*Reply: The authors agree that a broad and comprehensive comparison between the intricacies of results and parameters using an array of surface velocity methods would be valuable research, particularly towards the development of an ensemble-based surface velocity analysis. However, the scope of this initial paper focusing on explaining the methodology and evaluation of the CVSG system for adaptive streamflow monitoring is already quite extensive for a single publication.*

Figure 2: It would be helpful to annotate these images to include the region of interest used by each method for computation of discharge. For example, your results demonstrate that the CVSG dramatically outperforms other methods for the irrigation channel in NSW. I'd like to know where the STIV search lines were placed? What is the ROI for the SSIV processing?

*Reply: The authors thank the reviewer for this feedback, and agree that this is a valuable illustration to provide to improve Figure S1 in the supplementary material with an annotated image. We have added this image as a second panel for Figure S1 (generated through programming a procedural pattern to apply to the coordinates for each region of interest).*

**Modification of Figure S1:**

[Figure]

**Figure S1: Raw discharge measurements using different technologies along the length of the irrigation channel in NSW, Australia.**

Line 292: Perhaps reword to indicate "diffuse light" rather that "softly lit" -- The light diffusion leads to reductions in shadows, making this dataset a great test for these methods.

*Reply: The authors thank the reviewer for this suggestion, and agree that this is better phrasing. We have made this change in line with this suggestion on L292.*

**Modification of L292:** "This benchmark case study presents a favourably diffusely lit environment with visible surface rippling features across the full width of the cross-section, and a sky/vegetation reflective water surface."

Figure 3:

It would be helpful to indicate that the ADCP data are the near-surface cell values. Additionally, label the plot to indicate these are surface velocity profiles. Although this is indicated in the text, it should also be included in the caption and/or figure legend.

Additionally, panel D may benefit from also including a residual plot. It seems that there is visually a trend in the CVSG results of under-predicting lower and over-predicting higher Qs. Alternatively, linear trend-lines could be added to show whether this is the case or not.

*Reply: We thank the reviewer for these valuable suggestions for improving the manuscript. We have added the indication of the near-surface cell ADCP observations to both the figure legend and caption, as well as changing the y-axis for the surface velocity profiles to be labelled as such. The authors also thank the reviewer for their suggestion for Figure 3d, however the only significant trend present between lower and higher Qs is in the CVSG discharges applying the stereophotogrammetry estimated water levels, where this trend is already clearly observed and noted in the discharge rating of Figure 7 and L593-596.*

**Modification of Figure 3:**

[Figure]

**Figure 2: Detailed time point comparison raw and model fitted velocity measurements plotted with nearest surface ADCP measurement cells over the cross-sections at (a) Castor River, Ontario, Canada, (b) an irrigation channel in NSW, Australia, and (c) Tyenna River, Tasmania, Australia. (d) Correlation plot between the gauge rating and optically estimated discharges at comparison time points at Tyenna River, Tasmania, Australia, with the detailed comparison time point indicated. CVSG 5-second duration surface velocities shown for (a) Castor River, Ontario, and (b) the irrigation channel in NSW, Australia. CVSG 10-second duration surface velocities shown for (c, d) Tyenna River, Tasmania, Australia. Hydro-STIV velocity estimates outlined in black were automatically produced, whereas the estimates outlined in red were corrected to the Fourier result or manually corrected to reduce automatically overestimated velocities resulting from the higher frequency surface wave patterns or underestimated tracer-poor search lines.**

Line 479: This makes sense, because of potential errors in the determination of WSE from the stereo cameras.

*Reply: The authors thank the reviewer for their feedback on the reasonableness of this line.*

Line 500: Was this primarily caused by clear water? Relatedly, does the CVSG manage to see any of the bed through the water? Perhaps if so, there might be some value in attempting to extract bed geometry, assuming a suitable correction for refractive properties can be found.

*Reply: This is precisely our understanding of the data. We have previously experimented and found surprising success with the capability to use refractive correction in the reconstruction of the submerged cross-section under the clear water conditions at the Tyenna River site, but have yet to apply and evaluate this approach across further sites or develop automated logic for when this is suitable to apply, and how to adaptively integrate these submerged cross-section measurements into the site cross-section.*

Line 561: The ability to capture low flow (high clarity water) image velocimetry measurements accurately continues to be a substantial challenge.

*Reply: The authors agree with this comment by the reviewer, and welcome future research towards non-contact methodologies that specifically improve measurements under these conditions.*

Line 674: Based on the findings of this paper, I agree with the later concept that CVSG ca nhelp identify low-flow site suitability. I am less convinced that CVSG (or any other image velocimetry approach) will inform low flow conditions.

*Reply: We agree with this assessment of image velocimetry approaches in general, and share this experience. L674 does not seek to make any statement that CVSG can inform low flow conditions, rather this line describes a practical solution to this limitation by providing the lower flow ratings to CVSG through manual gaugings where flow conditions are generally also safer for personnel and equipment.*

Section 5: Conclusions: One thing not discussed in this paper is the errors associated with cross-sectional area. It seems that since the CVSG is able to extract stereographic elevations of the low flow channel (or better yet dry channel), that there should be a way to consider changes in cross-sectional area ratings. Maybe something for a future paper?

*Reply: The authors thank the reviewer for this recommendation. We have added this to L736 in the conclusions.*

**Added from L697:** "This work did not address errors associated with cross-sectional area changes and the capability of the CVSG system to extract stereophotogrammetry estimated elevations of the dry channel areas to inform changes to discharge ratings, which is recommended for future research using stereo imagery-based optical stream gauging approaches."

Line 706: add a period to the end of the code availabilty sentence.

*Reply: We thank the reviewer for highlighting this oversight. We have added a period as suggested.*

**Added to L706:** "Code not available at the time of publication."

In addition to my official refereed comments, I would express agreement with RC1's general comments. Although I share RC1's frustration with the lack of explicity and thoroughly detailing the CVSG algoritms. I think that there is a valid criticism here that the paper results would be hard to repeat or evaulate given the details provided.

If the authors can expand on the details such as to address RC1's comments, I believe that many of my original comments will also be addressed.

*Reply: We thank the reviewer for adding their feedback on the clarity in the repeatability of the methods detailed in the manuscript. The authors have made changes to the manuscript based on the valuable feedback provided by the reviewers towards clarifying all of the points of confusion surrounding how the methods have been developed and applied.*

---

## Author Comment (AC3)

**Reviewer 3**

**General**

This manuscript introduces a new stereo computer vision stream gauging (CVSG) system for monitoring streamflow in rivers. Compared to existing systems, the added value of such a contactless streamgauge measuring both water level and surface flow velocity comes from the camera calibration without ground control points and the adaptive estimation of the rating curve. While the originality of some features of the system is real, several important methods are not described in enough detail so they could be understood and reproduced, which in my opinion cannot be accepted in a scientific publication. Even with this lack of information, some concerns arise about some methods involved, especially the velocity distribution model, and the rating curve model and estimation.

*Reply: The authors thank the reviewer for their general feedback and comments about the real added value and originality offered by this work. We understand that some clarification was required within the manuscript in order for the reader to more clearly understand the procedure of the methods and study being presented, and have made changes in order to address areas of confusion or lacking in clarity. We believe that the concerns raised about some of the methods involved were resulting from this confusion that was evident in the specific points that were noted.*

**Specific points**

L47-50: the text here suggests that rating curves could not be established with acceptable uncertainty in natural waterways without artificial controls. This is not true, as many hydrometric stations demonstrate.

*Reply: We thank the reviewer for their perspective on these lines, but we do not agree that these lines make this suggestion, while the following lines do describe the usefulness of the development of discharge rating curves in natural waterways.*

L61: you should also mention surface velocity radars as a technology affordable for contactless streamflow monitoring stations (e.g. Khan et al. 2021, Uncertainty in Remote Sensing of Streams using Noncontact Radars, J of Hydrology). This s an efficient alternative to image-based systems that should be compared and discussed.

*Reply: The authors thank the reviewer for this valuable suggestion for improving the manuscript's provision of relevant contextual background information to the reader. As such, we have added a line and suggested citation to this paragraph at L61 strengthening the introduction.*

**Modification from L61:** "Therefore, non-contact and affordable solutions such as radar (Rahman Khan et al., 2021) or optical, offer the potential to overcome these challenges by measuring velocity and stage without in-situ sensors. Similar to one of the oldest manual methods to measure velocities in a waterway by measuring the displacements of surface floats over time, the passive optical measurement of surface velocities using relatively inexpensive camera systems has been an attractive approach to stream gauging (Dobriyal et al., 2017)."

L71: Other commercial image-based stations exist, e.g. the product sold by Tenevia, France.

*Reply: We thank the reviewer for raising the Tenevia example that we are aware of, however we do not preclude the existence of other commercial image-based stations in the text, and we do not feel*

*it necessary or appropriate to list all image-based stations that have been developed (this is not a review paper). The authors did not find peer-reviewed research detailing the methods, application, or evaluation of the Tenevia product, and hence we have opted for mentioning and citing the details of another commercially available product as a referenced example.*

L87: 'initial surveying and calibration of new sites' is not a strong limitation for monitoring stations, as it is a limited additional effort compared to the installation of the system. Autocalibration would be more decisive for portable streamgauging systems (eg smartphone applications) for which surveying is a problem.

*Reply: The authors thank the reviewer for their comment, however the perspective of the 'initial surveying and calibration of new sites' being of limited effort compared to the installation of the system is not shared by the authors. The installation of the system presented is not a significant effort, and the necessary additional equipment, cost, expertise, errors, and time required for initial surveying and calibration is a strong motivation for this research and development. Furthermore, the stream gauging solution presented is a portable stream gauging system in its own right (as it can be moved between established sites similarly to existing smartphone applications mentioned by the reviewer).*

L118: 40 m is a limited range for stage measurements in medium to large rivers. Then L155, a range of 2 to 10 m is mentioned, which is very limited. What range is the right one?

*Reply: We thank the reviewer for their comment, and note that this confusion was also evidenced in a previous reviewer's comment. The 40 m mentioned on L118 is relative to the position of the camera and referring to the system hardware limitation for estimating water levels using stereophotogrammetry requiring the water near edge to exist within the vision of the camera within a 40 m range from the camera's physical location, while the 10 m on L156 is describing the region of the water surface used for stereophotogrammetry water level estimation relative to the near bank interception with the water edge.*

Also, using an IMU may be too expensive for just the initial survey of a station with a fixed angle and position… what is the additional cost and weight/size of an IMU?

*Reply: The authors thank the reviewer for their concern regarding the cost, weight, and size of an IMU, however we there is no additional expense and negligible weight/size required as we use the IMU embedded in the ZED 2/2i stereo camera (Stereolabs Inc., San Francisco, CA, USA). With the embedded IMU, we also note that, the CVSG system does not just use this hardware for initial surveying with fixed angle and position, but recalculates this orientation with reference to the previously sensed orientation, and measures the stability of the camera with each video recording analysed.*

Fig 1 is a good summary of the system but there is not enough information in the text (L179-193 especially) to understand the methods in a reproducible way. At least the principles should be explained and underlying equations provided so that the manuscript can be published as a research paper.

*Reply: We thank the reviewer for their compliment on the graphical summary of the system provided by Figure 1, and critical feedback about the level of detail provided in the text towards reproducing the methods. While we believe that every component of the methods described in the identified text is now an accurate and clear description of the simple arithmetic operations performed on the results of the optical flow field estimation, the underlying principles of the equations and procedure have*

*been explained in great detail with the optical flow solution utilised from the cited optical flow literature now additionally added to the manuscript in this section.*

**Modification from L171:** "Shi et al. (2020) compared three established and widely applied optical flow techniques to breaking surges, noting the advantages of the Farneback algorithm for its relatively high accuracy and dense flow fields, as well as a lower sensitivity to noise with the converging iterative solution for the displacement vector, $d$, between a pair of images using quadratic polynomials following Eq. (1):

$$d(\mathrm{X_{im}}) = \left(\sum_{\Delta X_{im} \in I_{local}} w A^T A\right)^{-1} \sum_{\Delta X_{im} \in I_{local}} w A^T \Delta b_f \,, \tag{1}$$

where $I$ is the greyscale image with local neighbourhood regions denoted by $I_{local}$ using the image coordinates $x_{im}$ and $y_{im}$ to form $X_{im} = \begin{bmatrix} x_{im} \\ y_{im} \end{bmatrix}$, where the change in brightness between the corresponding pixels in the pairs of images are denoted $\Delta X_{im}$. Furthermore, $w$ is a weighting function over the local neighbourhood regions, while the polynomials are defined by $f(x_{im}, y_{im}) \cong a_1 + a_2 x_{im} + a_3 y_{im} + a_4 x_{im}^2 + a_5 y_{im}^2 + a_6 x_{im} y_{im}$ with $A = \begin{bmatrix} a_4 & \frac{a^6}{2} \\ \frac{a^6}{2} & a_5 \end{bmatrix}$, $b_f = \begin{bmatrix} a_2 \\ a_3 \end{bmatrix}$, and $c = a_1$.

The approach is a variational method combining the assumptions of local neighbourhood brightness intensity variation between frames the minimisation of an energy function assuming a slowly varying displacement field for locally smooth velocity gradients (Shah and Xuezhi, 2021)."

Eq. 1: what is the physical justification (or reference) of this velocity distribution model? Why not using existing models, eg the Froude-based models, cf. Fulford and Sauer (1986)? This exponential model does not look very physical.

*Reply: The authors appreciate the reviewer's questions and commentary; however, the physical justification and motivation is described in the text explaining Eq. (1) (now Eq. (2)). The exponential relationship from the boundary distance factor to the surface velocity profile is just a simple logarithmic relationship of the rearranged form -bx = ln(1-V_s/V_infty). Many alternative relationships and boundary distance factor transformations were tested against optically estimated surface velocity distribution observations at different sites, but the form presented here was the best generically fitting to the data. While the authors appreciate the suggestion of the reviewer, the models suggested cannot be used to simply and reliably fit surface velocity measurements in the transformed boundary distance domain of generic cross-sections.*

L220: are the alpha values in Hauet et al. (2018) local or cross-sectional averages? Large differences between local and average values have been reported by Welber et al. (2016, WRR) for instance. How do your values compare with their empirical values? And with theoretical models, cf. eg LeCoz et al. (2010)?

*Reply: While the alpha values are applied locally (naturally lending more weight in estimating depth average velocities from the surface velocities in deeper flows above the defined threshold), the overall effect is to smoothly transition the effective global alpha value from the lower bound to the upper bound dependent on the distribution of water depth across the cross-section. The default values we have applied (derived from the empirical work of Hauet et al. (2018)) fall within the reported interquartile range of Welber et al. (2016). Their local alpha estimations are supportive of a transition below (the default) 2 m water depth, where the spread in the local alpha estimations was seen to increase dramatically. However, Welber et al. (2016) also note that care must be taken with the pairing of the ADCP results that were applied in the local estimations. The relation with the*

*theoretical models of LeCoz et al. (2010) were considered already in Hauet et al. (2018), and agrees with the range of default values applied in this work using the conclusions of Hauet et al. (2018).*

L236: again, equations are needed here, but the sentence suggests that a single power equation Q=a(H-b)^c is used for modelling the rating curve. At most streamgages, a single segment is not enough to build a rating curve due to multiple controls. You should review and use more relevant rating curve models and estimation methods, in particular refer to the comparison of 7 methods by Kiang et al. (2018, WRR) and explain how your method compares with the methods recently proposed by several research groups, some of them being publicly available.

*Reply: The authors thank the reviewer for their suggestion and acknowledge that the reviewer has become confused by the wording of these lines, resulting in the reviewer coming to the wrong understanding of how the discharge ratings are fitted. As such, we have improved the clarity of the details provided from L234, particularly noting that the power law weighted fitting method has not been applied in this work. Instead, the more preferable (and default) configuration for the CVSG system has been utilised, effectively applying a standard signal filter to the range of discharge estimations calculated across the range of water levels observed in the learning surface velocity distribution. The outcome of this method is a linearly locally fit discharge rating, which is an approach supported by the systematic arguments presented in Fenton (2018).*

**Modification from L234:** "The learning discharge rating can be configured to either be generated from the range of discharge estimates by directly applying a locally fitted Savitzky-Golay signal filter (Savitzky and Golay, 1964) (using a filter window size of 0.05 m vertically with nearest boundaries and linear fitting) or fitting a power law weighted by the number of observations and the optical flow coverage measured at each 0.01 m water level increment. The latter power law weighted fitting method has not been applied here, as the Savitzky-Golay signal filter is chosen instead for the results presented in this work (considered by the authors to be the preferred default configuration for general application following the arguments of Fenton (2018))."

L350: was the system placed too low due to its limited sensing range? This is a very problematic limitation, in practice.

*Reply: It is not anticipated in the experience of the authors that stream gauging infrastructure can always be immune to all possible flooding levels. The system was placed above the historic record flood levels, however the authors did not and could not have reasonably predicted that the new record highest flood would occur during the first twelve months of site deployment. The authors estimate that the camera could have been secured higher on the pole on site (which would be expected to either slightly improve or reduce the quality of the stereophotogrammetry estimated water levels and optical flow estimations to some degree). In this view we do not see how this is a very problematic limitation, in practice, relative to the benefits of affording more monitoring sites without the large infrastructure required to tentatively guarantee equipment survival.*

L370-378: this paragraph belongs to conclusions, not to results. Please move it to Conclusions or remove.

*Reply: We thank the reviewer for this suggestion. We agree that we can move this paragraph (representing an overview of the results section) to a results overview section at the end of the results, but we do not agree that this paragraph is a good fit for the conclusions.*

Fig. 3b: STIV and DischargeLab velocity measurements are much higher than reference (ADCP) velocities (and than CVSG velocities) in the irrigation canal case. What is the cause for such large, unsual errors? L423: what are the HydroSTIV 'ambiguous results'?

Should manual determinations of the STI slopes be used, as often done in practice? Is there some operator effect? This should be clarified.

*Reply: The authors thank the reviewer for this feedback, and agree that the manuscript could be improved with explicit outlining of the STI slopes in Figure 3 that were required to be adjusted in order to reduce the automatically overestimated velocities resulting from the higher frequency surface wave patterns or underestimated tracer-poor search lines.* This modification has been added to Figure 3 to clarify the determination method that was able to be used for each of the STI slopes across the cross-sections with this additional detail added to the figure caption. *Advice was sought from the official support provider for the Hydro-STIV software package where they kindly provided their own distortion correction ensuring that the camera was correctly calibrated. They provided advice confirming the camera location would need to be moved to a more consistent section of the channel in order to measure the flow using Hydro-STIV in a more stable manner.*

**Modification of Figure 3:**

[Figure]

**Figure 1: Detailed time point comparison raw and model fitted velocity measurements plotted with nearest surface ADCP measurement cells over the cross-sections at (a) Castor River, Ontario, Canada, (b) an irrigation channel in NSW, Australia, and (c) Tyenna River, Tasmania, Australia. (d) Correlation plot between the gauge rating and optically estimated discharges at comparison time points at Tyenna River, Tasmania, Australia, with the detailed comparison time point indicated. CVSG 5-second duration surface velocities shown for (a) Castor River, Ontario, and (b) the irrigation channel in NSW, Australia. CVSG 10-second duration surface velocities shown for (c, d) Tyenna River, Tasmania, Australia. Hydro-STIV velocity estimates outlined in black were automatically produced, whereas the estimates outlined in red were corrected to the Fourier result or manually corrected to reduce automatically overestimated velocities resulting from the higher frequency surface wave patterns or underestimated tracer-poor search lines.**

This case also shows that the velocity distribution model is inaccurate for such complex case. Then, what is the value of fitting such a model instead of using the high-resolution velocity measurements? Why not using the model only for interpolating missing data in unmeasured areas? The CVSG error with model fit (+55% in table 3) is clearly unacceptable and calls for not using such a model fit.

*Reply: The authors thank the reviewer for this comment, however the intention of the manuscript was to apply the methods presented equally to each site and condition as exampled. We believe that it is important to clearly show the circumstances in which aspects of the methodology presented are not applicable, and as such we seek to highlight these negative results.*

L461-464: this argument is weak because rating shifts may have occurred during such a long period of time. Also, the huge scatter in Fig 4 may be due to the same cause (rating shifts).

*Reply: We agree with the reviewer's assessment of the data, noting that the measurements indicate overall shifts in the discharge rating over the longer time scales presented. We have edited L461 to make this point clear, as well as adding emphasis to the main point of the figure with regards to showing the results of CVSG in context with the manual gaugings over a more significant timespan.*

**Modification from L461:** "However, it is important to note that the variability in CVSG discharge estimates is minimal compared to the variation in manual gauging estimates from similar water levels since 1989. This variation in discharge estimates over time is often a function of cross section changes and subsequent ratings shifts. Relative differences are expected to be within the realm of uncertainty of the true discharge, particularly as the discharge has only been measured at this water level once in 1966, with measurements within 0.005 m occurring five times (most recently in 1989), and 37 measurements within 0.05 m (the two most recent occurring 2 years and 8 years prior to the time of this case study recording) (Figure 4)."

L507-508 and L520: my conclusion is that image-based stage measurement is a failure. Modern contactless gauges such as radar gauges are a much better option in terms of cost and accuracy. And they also work at night and in the fog, rain, etc.

*Reply: Thank you for your comment. We agree that the stereo camera-based stage measurement is the greatest source of error in the discharge measurements presented in this manuscript. We have been very clear about that (see line 537). However, the comparison of stereo image-based stage measurements against other approaches such as radar is outside the scope of this manuscript. We have added a line to the introduction about the application of radar gauges to L61 to improve the background information contextualisation provided to the reader. We have strong beliefs that the best tool to apply is dependent on the specific site to be monitored and the objectives/requirements of the monitoring to be undertaken.*

**Modification from L61:** "Therefore, non-contact and affordable solutions such as radar (Rahman Khan et al., 2021) or optical, offer the potential to overcome these challenges by measuring velocity and stage without in-situ sensors. Similar to one of the oldest manual methods to measure velocities in a waterway by measuring the displacements of surface floats over time, the passive optical measurement of surface velocities using relatively inexpensive camera systems has been an attractive approach to stream gauging (Dobriyal et al., 2017)."

L548-550: measurement improvement through real-time learning seems to hide some error compensation, since stage measurements are affected by substantial errors. This is a problem, as a wrong rating curve is certainly established to cope with stage errors specific to the CVSG system. Such biased rating curve could not be used with conventional, accurate stage records…

*Reply: We thank the reviewer for raising this point, however we are not sure how the reviewer suggests that errors are being compensated for while all evaluations of the discharge rating curve occur on the same gauged reference datum. As a result of this comment, we have identified an improvement in the clarity of these lines describing the results. It should be highlighted that the cause of the similar raw discharge estimation errors between the analysis using the stereophotogrammetry estimated water levels and the analysis using the gauge water levels owes to the timing of these errors occurring during flow events with poor surface velocity visibility for the raw measurements acting independently of any learning surface velocity distributions.* *We have updated these lines from L550 to more clearly explain the presented results.*

**Modification from L550:** "Interestingly, the magnitude of raw CVSG discharge estimation errors was remarkably similar between the remotely sensed and gauge water level cases due to the most significant errors in the raw measurements occurring during flow events with poor surface velocity visibility. In these cases, the learning surface velocity distribution fitted model demonstrated significant improvements to the raw optical measurements. Further to this, the reduced water level estimation noise when using the gauge water level (Figure S8b) displayed significantly reduced error in the CVSG learning discharge estimations converging much faster between the real-time and 4-month hindsight rating estimates."

L587-589: acquiring measurements much faster than conventional streamgauging techniques is indeed a critical advantage of such image-based (or radar-based) velocimetry monitoring systems. However, the advantage is not specific to the CVSG system proposed here.

*Reply: The authors thank the reviewer for their comment, and we agree with every aspect of the statement by the reviewer. While we discuss this more broadly in the discussion, the authors do not see how this discussion point would be appropriate at these lines in the results.*

L592-594: this argument can be discussed depending on the rating model assumed. Unlike the vague description of the rating method before, here it is suggested that several (piecewise?) power segments are used to compute the rating curve… Details and equations are definitely needed for clarification. And the 'smoother fit of the gauging station rating curve' is not necessarily less accurate than a more flexible rating curve model because it usually rely on physically-based considerations, ensuring a better extrapolation for high flows, for instance.

*Reply: We thank the reviewer for their feedback, and note that this confusion in the assumption of the rating curve has been addressed from a prior comment from the reviewer.*

More generally, it is a pity that no uncertainty intervals around the rating curve estimates are presented, whereas methods are ow available for this (cf. Kiang et al. 2018 and the associated methods). Accounting for the variable uncertainty of discharge measurements is especially important for surface velocity methods like the CVSG.

*Reply: We thank the reviewer for this feedback, and note that similar comments were submitted by a previous reviewer. The authors have decided to leave out the "envelope" results representing a form of uncertainty bounds (which are generally enveloping of all available discharge estimation technologies) to avoid cluttering the results, and focus on the differences between the best estimate provided (particularly given that a single number is ordinarily taken as the best estimate from the gauging stations compared). The authors agree with the reviewer around the need for proper uncertainty estimates, however we consider what constitutes a 'proper' uncertainty estimate to be reasonably debatable, and the authors do not yet seek to present techniques or support of any particular set of uncertainty estimation approach in connection with this work beyond the methodology detailed. The key points in the source mentioned by the reviewer reinforce the wide*

*variety of uncertainty estimates possible using different methods, requiring careful understanding of the assumptions behind the uncertainty methods used for interpreting the results of any uncertainty estimations provided.* *We have added a line in the discussion at L636 on this point with reference to the reviewer's suggested citation.*

**Modification from L636:** "Alternative methods for estimating the uncertainty of stream discharge rating curves have been compared in Kiang et al. (2018), finding a wide variation in uncertainty estimates resulting from different methods which demonstrated the necessary careful selection and communication of the assumptions of the uncertainty estimates provided."

The Section 4 'Discussion' needs to be more formally organized around precise questions to be more precisely related to the methods and results of the paper. Also, more references should be used, in particular on surface velocity radar stations and index velocity methods (as an alternative to image-based streamflow monitoring stations), rating curve estimation methods (including the modern data assimilation methods already mentioned, cf. Kiang et al.), other image-based monitoring solutions (e.g. Tenevia video stations, and Stumpf et al. (2016, WRR) is a needed reference on stereo cameras L668).

*Reply: We thank the reviewer for their suggestion for more formally organising the discussion section around precise questions.* *We have added a line to the discussion referencing the valuable additional citation suggested by the reviewer (Kiang et al., 2018) for strengthening the manuscript at L636 (a modification from the previous comment). We have also added a reference in the discussion to Stumpf et al. (2016) which applied and evaluated a photogrammetry technique for measuring water level and discharge using cameras with different perspectives at L667 (now L730).* *However, we do not feel it is appropriate to discuss surface velocity radar stations which did not form any part or comparison in the study, just as we do not discuss every measurement technique beyond the scope of this work. Further to this, we do not believe restructuring the discussion will improve the readability, as the ideas in the discussion have been structured already with the intention of providing the reader with a logical order and natural flow.*

**Added to L667:** "Significant work has been undertaken towards developing and applying photogrammetry techniques operating using different camera perspectives from more than one camera for long-term automated water level and discharge measurements (Stumpf et al., 2016)."

The Section 5 'Conclusions' does not provide a real summary of the results, including success and failure of the attempts. It thus fails to present perspectives for improving or extending the system. The first sentence (L693) is highly questionable as the study does not demonstrate the 'successful development' of the system since at least some parts of the methods have failed or could not be tested, including the stage measurements, the velocity distribution model, the night measurements, etc.

*Reply: The authors thank the reviewer for their feedback on the conclusions section.* *We have removed the word successful from the conclusions and added a sentence highlighting the specific challenges that remain to be addressed through future work.*

**Modification from L693:** "This study has demonstrated the development of an automated operational optical stream gauging system employing methods providing improved reliability for remotely gauging streams using state-of-the-art surface velocimetry technologies across varying flow and lighting conditions. Evaluation of the existing best practice in available stream measurement technologies and published discharge ratings across the array of site conditions evident in this work demonstrated that the methods in this study were similarly effective for gauging stream discharge to existing accuracy benchmarks. This work did not address errors associated with cross-sectional area changes and the capability of the CVSG system to extract stereophotogrammetry estimated elevations of the dry channel areas to inform changes to discharge ratings, which is recommended for future

research using stereo imagery-based optical stream gauging approaches. In addition, the challenges associated with analysing surface velocity at night and quantifying water level through stereophotogrammetry under a range of lighting conditions and greater distances provide opportunities for future work. Despite these challenges, non-contact and automated solutions offer a significantly greater density of velocity-stage observations resulting in up-to-date adaptively learning discharge ratings through time. As climate-driven extreme weather events increase in frequency, it is increasingly important to develop and apply flexible monitoring tools, such as CVSG, that can reduce the human and environmental risks associated with traditional approaches and deliver real-time data to water resource managers."

**Minor points**

Abstract L21: 'error margins of 5-15%', what do you mean precisely? Is this the uncertainty at a given probability level? Or what?

*Reply: The authors thank the reviewer for raising this question relating to this line in the abstract. We had intended the reference to the 'within the best available measurement error margins of 5-15%' to refer to the general range of results between the best available measurement approaches which were evaluated in this study.* *We have updated this line at L21 to better clarify the meaning of this general result summarised in the abstract.*

**Modifications from L18:** "Evaluations between reference state-of-the-art discharge measurement technologies using DischargeLab (using surface structure image velocimetry), Hydro-STIV (using space-time image velocimetry), ADCPs (acoustic doppler current profilers), and gauging station discharge ratings demonstrated that the optical surface velocimetry methods were capable of estimating discharge within a 5-15% range between these best available measurement approaches."

L46: Doppler

*Reply: We thank the reviewer for highlighting this oversight.* *We have corrected the proper capitalisation of Doppler.*

**Modification from L41:** "Intrusive methods range from the resource intensive installation of hydraulic control structures to measure discharge rates analytically using simpler water level measurements within a designed range by obstructing and controlling the flow through a standardised geometry (Boiten, 2002) (often to the detriment of aquatic species (Mueller et al., 2011), as well as sedimentation and erosion (Pagliara and Palermo, 2015; Ogden et al., 2011)), through to the risking of people and equipment entering the stream to measure velocities using passive mechanical current meters or active acoustic Doppler velocimetry profiles (Gordon, 1989)."

L526: true dischargeS

*Reply: We thank the reviewer for highlighting this grammatical point.* *We have corrected the pluralisation of 'discharge' in this line.*

**Modification from L526:** "Even though the true discharges at the measurement times are not known, the CVSG learning discharge estimations using the gauge water levels at the time overestimated the discharges of events occurring in April 2021 relative to the latest gauging station discharge rating by up to 20%."

L527: 'somewhat overestimated': this is vague, by how much?

*Reply: The authors thank the reviewer for raising this minor point, and agree that a more specific description of the amount of overestimation would improve the manuscript.* *As such, we have removed the vague descriptor, 'somewhat', and added to the end of the line 'by up to 20%'. The modification of this line is included in the modification for the previous comment.*

Fig 6 caption: 'and gauge water levels', remove 'and'

*Reply: We thank the reviewer for highlighting this repeated 'and' in the figure caption. We have removed this redundant 'and'.*

*Modification from Figure 6 caption:*

**Figure 2: Correlation plots for the latest gauging station rating discharge timeseries against the CVSG estimated discharge timeseries at Tyenna River, Tasmania, Australia using (a) stereophotogrammetry estimated water levels, and (b) gauge water levels, as well as at Paterson River, NSW, Australia using (c) stereophotogrammetry estimated water levels, and (d) gauge water levels.**